# Quantifying Generalization Complexity for Large Language Models

**Zhenting Qi**[1,*] **Hongyin Luo**[2]**, Xuliang Huang**[3]**, Zhuokai Zhao**[4,5]**, Yibo Jiang**[5]
**Xiangjun Fan**[4]**, Himabindu Lakkaraju**[1]**, James Glass**[2]
[1]Harvard University, [2]Massachusetts Institute of Technology
[3]University of Illinois at Urbana-Champaign, [4]Meta, [5]University of Chicago
zhentingqi@g.harvard.edu, hyluo@mit.edu

## Abstract

While large language models (LLMs) have shown exceptional capabilities in understanding complex queries and performing sophisticated tasks, their generalization abilities are often deeply entangled with memorization, necessitating more precise evaluation. To address this challenge, we introduce **Scylla**, a dynamic evaluation framework that quantitatively measures the generalization abilities of LLMs. Scylla disentangles generalization from memorization via assessing model performance on both in-distribution (ID) and out-of-distribution (OOD) data through 20 tasks across 5 levels of complexity. Through extensive experiments, we uncover a non-monotonic relationship between task complexity and the performance gap between ID and OOD data, which we term the *generalization valley*. Specifically, this phenomenon reveals a critical threshold—referred to as *critical complexity*—where reliance on non-generalizable behavior peaks, indicating the upper bound of LLMs' generalization capabilities. As model size increases, the critical complexity shifts toward higher levels of task complexity, suggesting that larger models can handle more complex reasoning tasks before over-relying on memorization. Leveraging Scylla and the concept of critical complexity, we benchmark 28 LLMs including both open-sourced models such as LLaMA and Qwen families, and closed-sourced models like Claude and GPT, providing a more robust evaluation and establishing a clearer understanding of LLMs' generalization capabilities.

## 1 Introduction

Large language models (LLMs) have revolutionized natural language processing by exhibiting exceptional abilities in understanding complex queries, generating human-like text, and performing a variety of downstream tasks (OpenAI, 2023; Google, 2024; Bubeck et al., 2023; Hoffmann et al., 2022). Beyond their impressive text-generation capabilities, these models also demonstrate emerging skills in *reasoning* (Wei et al., 2022b; Kojima et al., 2022). Through increased inference-time computation (Chen et al., 2024b; Snell et al., 2024; Bansal et al., 2024; Qi et al., 2024; Wang et al., 2024), LLMs have achieved or even surpassed human-level performance on benchmarks that require nontrivial reasoning abilities (Cobbe et al., 2021; Hendrycks et al., 2021; 2020; Chen et al., 2021; Han et al., 2022). Despite these impressive advancements, research has also demonstrated that LLMs face significant challenges when solving problems that involve terms, patterns, or concepts that are less common in their training data (Razeghi et al., 2022; Kandpal et al., 2023; Chen et al., 2024a; Antoniades et al., 2024). Additionally, concerns have been raised regarding *data contamination* (Magar & Schwartz, 2022b; Carlini et al., 2022; Dong et al., 2024), as many benchmark datasets are sourced from the web and may overlap with the training data, either directly or indirectly, which undermines the reliability of results on such benchmarks. Consequently, there is ongoing debate about whether LLMs truly possess human-like reasoning abilities or simply rely on memorized patterns when solving problems (Kambhampati, 2024; Schwarzschild et al., 2024).

---

[*]Work completed during Zhenting's visit to MIT CSAIL. Source code will be available at https://github.com/zhentingqi/scylla.

Several efforts have been made to explore the interplay between generalization and memorization in LLMs' reasoning behaviors (Wu et al., 2023; Lotfi et al., 2023; Zhu et al., 2023; Antoniades et al., 2024; Dong et al., 2024). Lotfi et al. (2023) present the first non-vacuous generalization bounds for LLMs, demonstrating their ability to discover patterns that generalize to unseen data. Wu et al. (2023) suggest that generalization and memorization often exist on a continuum, as LLMs exhibit above-random performance on counterfactual tasks, though with some degradation compared to default tasks. Their study proposes that the seemingly "reasoning" behaviors of LLMs may stem from a combination of: (1) *generalization behaviors*, such as abstract logic and learned skills, and (2) *memorization behaviors*, including memorized input-output mappings and pattern matching. Despite these recent insights, the relationship between task difficulty, model size, and the balance between generalization and memorization remains poorly understood. Several factors undermine the robustness of current findings. First, reliable methods for quantifying task difficulty are still underdeveloped, and the distinction between problem length and intrinsic task complexity is often overlooked. Additionally, evaluations are usually complicated by data contamination and the entanglement with knowledge, introducing confounding factors to reasoning assessments.

In this work, we quantify the generalization ability of LLMs by aligning models with the intrinsic complexity of reasoning tasks. We address two specific research questions: 1) *How does task complexity affect the balance between generalizable (generalization) and non-generalizable (memorization) behaviors?* 2) *How does model size influence this balance?* We first develop a novel evaluation framework, SCYLLA, that is **sc**alable in task complexity, d**y**namic, know**l**edge-**l**ight, and memorization-**a**ware. We explain the necessity of each of these criteria for understanding the working mechanism of generalization, and show that no existing evaluation methods fully meet them. SCYLLA enables the generation of in-distribution (ID) and out-of-distribution (OOD) data of a given task, and the performance gap between them is considered an indicator of reliance on non-generalizable behaviors to solve the task. This allows us to assess how well models generalize learned task skills beyond their training distribution. We evaluate the performance of LLMs on both ID and OOD data across varying levels of quantified task complexity. The results of our experiments lead to two key findings:

1) **Non-monotonic performance gap across task complexity:** the performance gap between ID and OOD data initially widens as task complexity increases, reaches a peak, and then narrows as tasks become more complex—a phenomenon we refer to as *generalization valley*. As shown in Fig. 1, this non-monotonic relationship suggests that LMs are most vulnerable to distribution shifts at certain intermediate task complexity levels, where overfitting to training data leads to a greater dependence on memorization. and 2) **Peak of generalization valley shifts rightward with increasing model size:** as the model size increases, the peak of performance gap, referred to as the *critical complexity*, shifts to the right. As shown in Fig. 1, this rightward shift indicates that larger models are better equipped to handle more complex tasks without over-relying on memorization, maintaining generalization capabilities across a broader range of task difficulties.

The contributions of this paper are fourfold. First, we present a novel, task-centric evalua-

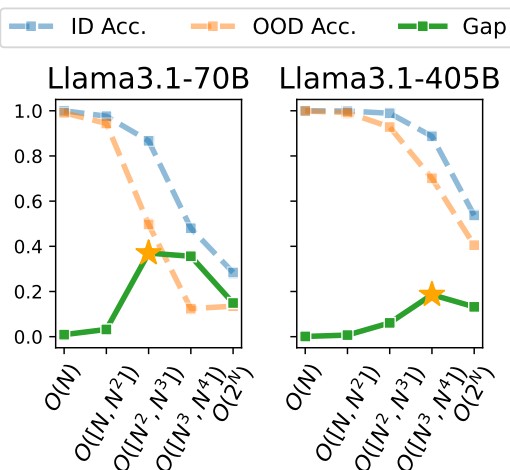

Figure 1: An illustration of *generalization valley*, where the reliance on non-generalizable behaviors first increases and then decreases; and *critical complexity shift*, where the peak of the valley shifts rightward as model size increases.

tion framework that is scalable in task complexity, dynamic, knowledge-light, and memorization-aware, specifically designed to overcome limitations found in existing evaluation methods. Second, through a detailed analysis of performance across varying task complexities and model sizes, we uncover insights into generalization behavior, revealing patterns that distinguish when models increasingly rely on memorization versus generalization. And third, we highlight the impact of model size on generalization, demonstrating that larger models experience a delayed over-reliance on memorization, with peak performance discrepancies occurring at higher task difficulties than in smaller

models. Finally, leveraging our proposed framework and insights, we define a new metric that aims to reward models with strong generalization to OOD data while penalizing those that exhibit overfitting to ID data, and conduct a comprehensive benchmarking of 28 popular LLMs, focusing on their genuine reasoning capabilities.

## 2 RELATED WORK

### 2.1 GENERALIZATION & MEMORIZATION IN LLMS' REASONING

The debate over whether LLMs can genuinely reason or simply rely on memorization remains central to understand their true capabilities (Zhang et al., 2021; Tänzer et al., 2021; Zečević et al., 2023; Tang et al., 2023; Yin et al., 2023; Biderman et al., 2024). Wu et al. (2023) argue that models like GPT-4 perform well on default tasks but struggle significantly with counterfactual ones, implying that much of their success comes from memorization of specific patterns. Similarly, Dong et al. (2024) highlight how data contamination can inflate perceived generalization by enabling models to rely on memorized data, while Antoniades et al. (2024) observe that even larger models, which generally show stronger generalization, still exhibit memorization behaviors, especially for frequently encountered n-grams. Kambhampati (2024) assert that LLMs primarily perform "approximate retrieval" from large pretraining datasets rather than true reasoning. In contrast, Lotfi et al. (2023) introduce the first non-vacuous generalization bounds for LLMs, providing a mathematical framework that demonstrates LLMs' capability to discover regularities and generalize beyond their training data, particularly as models scale up, and thus disprove that larger LLMs are simply better at regurgitating training data. Together, these works highlight the ongoing tension between memorization and generalization, and the need for more robust evaluations that differentiate between the two.

### 2.2 EVALUATION OF LLMS' REASONING ABILITIES

Reasoning is recognized as a key component of both human cognition and AI development, driving research to evaluate the reasoning abilities of LLMs (Zhu et al., 2023). Recent research has emphasized tasks requiring logic and deduction—such as those involving math, text, and code—as benchmarks for reasoning across domains, generally divided into static and dynamic categories. Specifically, static benchmarks, including MATH (Hendrycks et al., 2021), GSM8K (Cobbe et al., 2021), BoardgameQA (Kazemi et al., 2023), and FOLIO (Han et al., 2022), use mathematical and logical problems to assess reasoning performance. However, these benchmarks, which remain fixed after publication, are vulnerable to data contamination (Magar & Schwartz, 2022a; Golchin & Surdeanu, 2023) and reasoning gap issues (Srivastava et al., 2024). To address these limitations, recent benchmarks have adopted a dynamic approach, either by generating new problem instances at test time or by regularly refreshing test data. CLRS-Text (Veličković et al., 2022), for instance, draws on algorithms from Introduction to Algorithms (Cormen et al., 2009) and synthesizes algorithmic reasoning problems in text form. Similarly, NPHardEval (Fan et al., 2023) is built upon algorithm tasks. It organizes them by complexity class, defines difficulty levels by problem lengths, and refreshes data on a monthly basis to mitigate the risk of overfitting. LiveBench (White et al., 2024) also frequently updates questions from the most recent information sources, but the task set tends to be too knowledge-intensive to be a good testbed for reasoning. DyVal (Zhu et al., 2023) employs a graph-informed algorithm to generate math, logic, and algorithm test cases, but faces challenges in manually specifying problems as graphs and defining valid constraints. Despite these research efforts, they barely disentangle reasoning and generalization from memorization or quantitatively align the intrinsic task complexity and generalization ability.

## 3 METHOD

### 3.1 MOTIVATION

To conduct reliable evaluations of the generalization capabilities of LLMs, we begin by discussing several key features that are essential for an effective benchmark. While prior research has touched upon the importance of scalable and dynamic evaluation (Zhu et al., 2023; Fan et al., 2023), we refine these criteria with clearer definitions and emphasize two additional critical dimensions: knowledge-light and memorization-aware.

**Scalable inherent complexity.** The difficulty of an ideal evaluation task should be both quantifiable and scalable (Zhu et al., 2023; Fan et al., 2023). There are two dimensions that influence the difficulty of solving a task: (1) the intrinsic complexity of the task and (2) the length of an input problem instance. The former refers to tasks that inherently require more sophisticated reasoning and a greater number of intermediate steps, with the number of steps increasing as a function of the length of the input instances. Consequently, when benchmarks increase task difficulty by both extending input lengths and introducing tasks of varying complexity, as seen in (Fan et al., 2023; Zhu et al., 2023), it becomes difficult to distinguish whether performance drops stem from the challenges with longer inputs or the tasks' intrinsic complexity. Plus, LLMs are known to struggle with length generalization problem, exhibiting a sharp decline in performance (Anil et al., 2022) or becoming unstable (Zhou et al., 2024) as input length increases. For these reasons, problem length is not an ideal hyperparameter for adjusting task difficulty and should be controlled. In other words, task difficulty should scale independently of input length.

**Dynamic question generation.** Optimal benchmark task instances should be generated dynamically to minimize the risk of data contamination. Many widely adopted reasoning benchmarks, such as those for mathematical reasoning (Cobbe et al., 2021; Hendrycks et al., 2021), are based on static datasets. However, evaluations based on static data often encounter challenges such as the reasoning gap problem (Srivastava et al., 2024) and data contamination (Magar & Schwartz, 2022a; Golchin & Surdeanu, 2023) issues, reducing the robustness and reliability of these assessments.

**Knowledge-light prerequisite.** Ideal evaluation tasks for reasoning should require minimal background knowledge, containing only simple task descriptions and queries. By minimizing the reliance on external information, we ensure that any performance differences is most likely attributable to the models' reasoning abilities rather than disparities in their knowledge bases, eliminating the ambiguity of whether a model's failure is due to a lack of necessary knowledge (Kandpal et al., 2023; Srivastava et al., 2022; Suzgun et al., 2022) or an inherent limitation in reasoning ability.

**Memorization-aware evaluation.** The benchmark should explicitly differentiate between task instances that are more likely to have been memorized and those that are less likely. This differentiation helps us accurately attribute the model's performance to either memorization or generalization.

## 3.2 SCYLLA BENCHMARK

Considering that none of the current benchmarks or evaluation frameworks meet all of the requirements defined above, we propose a new benchmark, SCYLLA, which distinguishes itself from existing benchmarks through the following key features:

- **S**calable inherent **C**omplexity: We utilize algorithmic complexity to quantify task complexity, defining tasks as more complex when they require algorithms of higher complexity for their solution. To ensure consistent complexity across tasks, we impose explicit constraints on problem lengths, ensuring the variation remains within a stable range while keeping the upper bound manageable, so that task complexity is minimally influenced by problem length. Our choice of tasks and their corresponding complexity bounds are detailed in §3.2.1.

- **DY**namic problem generation: All data are generated during the evaluation, ensuring that each evaluation instance is unique and unaffected by pre-exposed data. Details of the data synthesis methodology can be found in §3.2.2.

- Know**L**edge-**L**ight prerequisite: Tasks are designed to require no background knowledge, featuring simple and clear descriptions and straightforward instructions. All the tasks are designed to be solvable with basic skills such as additions and comparisons of non-negative integers.

- Memorization-**A**ware evaluation: Generalization and reasoning capabilities are more clearly disentangled from memorization through explicit differentiation between in-distribution (ID) and out-of-distribution (OOD) data. Performance on ID data reflects a combination of both generalization and memorization, as the model is familiar with both the length and patterns of the task instances. Conversely, performance on OOD data primarily indicates generalization, as the model is only familiar with the length of the task instances but not the specific patterns of the task elements. Therefore, we propose to utilize the performance gap between ID and OOD data as an estimation of the model's reliance on memorization, allowing us to assess how well models generalize learned task skills beyond their familiar instances. Procedures for generating ID and OOD data are detailed in §3.2.2.

Additionally, SCYLLA requires only black-box access to an LLM, enabling users to evaluate and compare different models across platforms without delving into their internal workings. Its task-centric design also ensures high adaptability and extensibility, allowing users to seamlessly customize and expand it by incorporating new tasks and complexity levels to make the evaluation results more precise and reliable. Further comparisons with existing benchmarks or evaluation frameworks can be found in Appendix A.2.

### 3.2.1 TASKS & COMPLEXITY LEVELS

To categorize and define the tasks in our benchmark, we adopt the notion of time complexity, which measures the order of growth of an algorithm's running time and provides a standardized framework for comparing the efficiency of different algorithms (Cormen et al., 2009). In this context, we *re-purpose LLMs as algorithm executors*, where their ability to handle tasks is expected to depend on the underlying computational complexity of those tasks. Our experiments, detailed later, validate this hypothesis by demonstrating that time complexity provides a meaningful and rigorous method for task classification.

As shown in Fig. 2, we first introduce **anchor tasks** that define the intrinsic complexity levels in our benchmark. Specifically, we utilize six levels of time complexity: $O(N)$, $O(N \log N)$, $O(N^2)$, $O(N^3)$, $O(N^4)$, and $O(2^N)$ to define our complexity intervals. For a specific task, we define its difficulty using a complexity range, denoted as $O([C_1, C_2])$, where $C_1$ and $C_2$ represent the lower and upper bounds of the time complexity for algorithms that solve the task. The anchor tasks are selected based on two key criteria. First, the time complexity of the selected tasks should fall within one or two adjacent complexity levels to maintain consistency in difficulty. This ensures that the tasks within each group are comparable in terms of computational demands, preventing significant changes in difficulty. It also avoids scenarios where models face both very simple and highly complex tasks, which could lead to inconsistent performance measurements and obscure the underlying reasoning capabilities we aim to evaluate. Second, the tasks must be simple and should avoid reliance on advanced mathematical knowledge, common sense, or natural language understanding—factors outside the scope of the reasoning abilities we aim to evaluate. For instance, matrix multiplication involves specific mathematical concepts, which are not aligned with our focus on reasoning capabilities. In contrast, tasks like "find max", which only require basic number ordering, provide a more appropriate measure of reasoning ability.

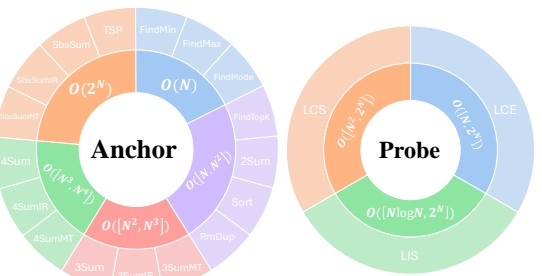

Figure 2: **Left**: Anchor tasks. These tasks form the core of our benchmark, providing a structured set of challenges across varying time complexities. **Right**: Probe tasks. These tasks are used to evaluate the level of complexity that LLMs adopt to solve them.

For anchor tasks, we propose three to four tasks for each complexity interval. The time complexities of these tasks are illustrated in the left of Fig. 2, with complexity increasing in a clockwise direction on the pie chart. Some task names are abbreviated in the figure, but detailed in Table 1.

To explore how LLMs handle tasks with multiple solutions of varying time complexities, we also introduce a set of tasks termed **probe tasks**. These tasks, including longest common subarray (LCS), longest increasing sequence (LIS), and longest consecutive elements (LCE), allow us to

Table 1: Anchor tasks for each complexity interval.

| | |
|---|---|
| $O(N)$ | Find min, find max, find mode |
| $O([N, N^2))$ | Find top-k, two sum, sort numbers, remove duplicates |
| $O([N^2, N^3))$ | 3-sum multiple ten, 3-sum in range, 3-sum |
| $O([N^3, N^4))$ | 4-sum multiple ten, 4-sum in range, 4-sum |
| $O(2^N)$ | Subset sum multiple ten, subset sum in range, subset sum, travelling salesman problem |

observe whether LLMs favor more efficient solutions as task complexity increases. The corresponding time complexities of these tasks are shown in the graph on the right of Fig. 2. The behavior of LLMs in choosing between multiple solutions for these tasks is further discussed in §4.5. For both the anchor tasks and probe tasks, a detailed explanation of their time complexity, along with other relevant information, can be found in Appendix D.

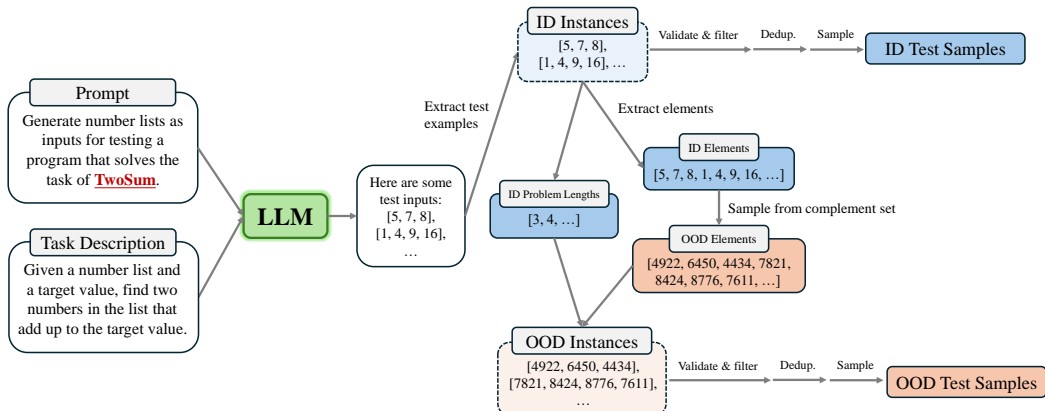

Figure 3: Pipeline for generating ID and OOD dataset for a given task, tailored to each LLM.

Our time complexity-based approach contrasts with benchmarks such as CLRS (Veličković et al., 2022; Markeeva et al., 2024) by not only grouping tasks by algorithm type, but also systematically representing the relationships between tasks of varying complexities. While approaches like NPHardEval (Fan et al., 2023) and DyVal (Zhu et al., 2023) categorize tasks based on input length, they lack a clear connection between input length and task difficulty, and thus makes it difficult to discern whether a lower performance stems from handling larger inputs or from the higher intrinsic complexity of the task itself. In contrast, SCYLLA improves upon these approaches by following more rigorous procedures of selecting tasks, defining complexities, and generating data, providing more reliable evaluations of LLMs' generalization abilities.

### 3.2.2 DATA SYNTHESIS PIPELINE

The entire pipeline for synthesizing data is illustrated in Fig. 3. For a given task, the first step is to obtain the ID test data, consisting of instances that the LLM is already familiar with. However, directly accessing the pre-training dataset of an LLM is often impractical, as the dataset composition and pre-training mixtures are generally not publicly available. Moreover, the vast size of the text corpora makes processing challenging. In addition, proprietary models like GPT-4 usually provide only black-box APIs, limiting the ability to conduct in-depth analysis of the model itself.

To address these challenges, we propose a workaround to approximate the ID data by directly querying the model about the distribution it is familiar with. Specifically, we first prompt the model to generate a substantial amount of test inputs tailored to the task description (see Appendix D.6 for prompt details). From these generated responses, we use regular expressions to extract test examples and designate them as the *ID test data*. Additionally, the individual numbers within these examples are identified as *ID elements*, with their lengths referred to as *ID problem lengths*. In our experiments, we typically prompt models to generate no less than 10k instances and obtain around 100k ID elements for each task.

In Fig. 4, we show an example where we query Mistral 7B model on the task of *Find Longest Increasing Subsequence*. Besides this task, we observe that for tasks in SCYLLA, most of the ID elements fall within the range of (0, 100), and the example lengths are typically between 4 and 16. For further validation of this approximate method, please refer to Appendix B. Next, we sample *OOD elements* from the complement of the ID elements, *i.e.*, the set of elements that do not overlap with the ID elements. This is implemented by sampling elements from a univer-

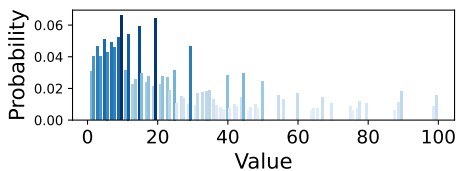

Figure 4: An example of probability distribution of ID elements collected by querying Mistral 7B v0.3 on the task of *Find Longest Increasing Subsequence*. The histogram shows the most frequent values with probabilities adding up to 90% (top-p=0.9).

sal set and excluding those that overlap with the ID elements. To identify an appropriate universal set, one can manually check the distribution of the extracted ID elements, as shown in Fig. 4. From our experiments, a large and diverse range of numbers like [0, 9,999] is sufficient for defining the universal set for all the LLMs we have tested. Note that the upper bound of the universal set is also

| Acc. on ID data (Mean ± Std. Dev.) | | | | | | |
| --- | --- | --- | --- | --- | --- | --- |
| $O(N)$ | 0.78 ± 0.12 | 0.8 ± 0.18 | 0.91 ± 0.07 | 0.95 ± 0.04 | 0.99 ± 0.02 | 0.95 ± 0.05 | 0.99 ± 0.01 |
| $O([N, N^2])$ | 0.45 ± 0.31 | 0.32 ± 0.2 | 0.53 ± 0.19 | 0.75 ± 0.23 | 0.83 ± 0.19 | 0.95 ± 0.05 | 0.96 ± 0.03 |
| $O([N^2, N^3])$ | 0.05 ± 0.03 | 0.11 ± 0.1 | 0.13 ± 0.07 | 0.19 ± 0.1 | 0.3 ± 0.05 | 0.26 ± 0.19 | 0.36 ± 0.03 |
| $O([N^3, N^4])$ | 0.03 ± 0.04 | 0.06 ± 0.06 | 0.1 ± 0.06 | 0.06 ± 0.04 | 0.21 ± 0.1 | 0.22 ± 0.14 | 0.36 ± 0.22 |
| $O(2^N)$ | 0.05 ± 0.08 | 0.05 ± 0.06 | 0.09 ± 0.15 | 0.05 ± 0.06 | 0.07 ± 0.04 | 0.06 ± 0.03 | 0.18 ± 0.12 |

| Acc. on OOD data (Mean ± Std. Dev.) | | | | | | |
| --- | --- | --- | --- | --- | --- | --- |
| $O(N)$ | 0.66 ± 0.21 | 0.72 ± 0.13 | 0.91 ± 0.03 | 0.9 ± 0.07 | 0.98 ± 0.02 | 0.93 ± 0.07 | 0.98 ± 0.03 |
| $O([N, N^2])$ | 0.34 ± 0.19 | 0.26 ± 0.17 | 0.41 ± 0.21 | 0.67 ± 0.14 | 0.67 ± 0.35 | 0.89 ± 0.16 | 0.96 ± 0.03 |
| $O([N^2, N^3])$ | 0.05 ± 0.03 | 0.12 ± 0.04 | 0.16 ± 0.07 | 0.13 ± 0.09 | 0.22 ± 0.1 | 0.19 ± 0.11 | 0.25 ± 0.09 |
| $O([N^3, N^4])$ | 0.01 ± 0.0 | 0.07 ± 0.04 | 0.09 ± 0.03 | 0.04 ± 0.03 | 0.16 ± 0.11 | 0.09 ± 0.07 | 0.21 ± 0.02 |
| $O(2^N)$ | 0.03 ± 0.07 | 0.03 ± 0.03 | 0.07 ± 0.13 | 0.05 ± 0.06 | 0.06 ± 0.04 | 0.03 ± 0.01 | 0.1 ± 0.13 |
| | 1.8B | 4B | 7B | 14B | 32B | 72B | 110B |

Figure 5: ID/OOD performance of Qwen 1.5 family across five complexity levels.

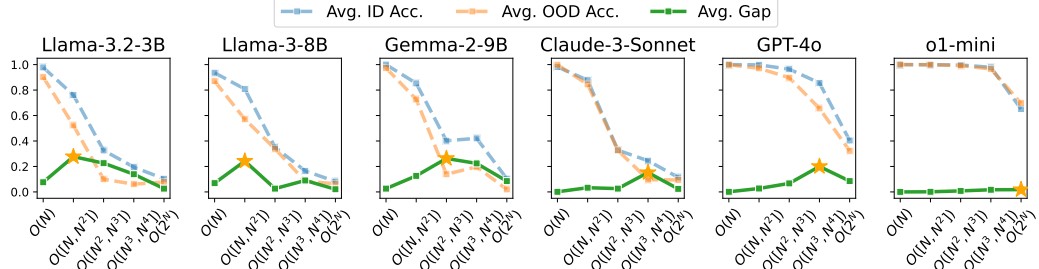

Figure 6: ID and OOD accuracy and performance gap curves for Llama-3.2-3B, Llama-3-8B, Gemma-2-9B, Claude-3-Sonnet, GPT-4o, and o1-mini. A significant drop in OOD accuracy is observed at the models' critical complexity, indicating a sudden decline in generalization ability.

restricted to avoid too many extra tokenizations of large numbers. To control for the length factor, we use the ID example sizes and OOD elements to construct OOD examples and obtain the *OOD test data*. During the actual benchmark experiments, we notice that some generated examples may be invalid or duplicated for both ID and OOD test data. To resolve this problem, we apply a validation process, which filters out invalid examples and removes duplicates. Finally, 256 test samples are selected for each of the ID and OOD dataset.

## 4 EXPERIMENTS

In this section, we conduct extensive experiments using SCYLLA on a series of LLMs. Throughout §4.1, §4.2 and §4.4, we use **anchor** tasks to conduct evaluations. In §4.5 we use **probe** tasks to examine whether LLMs favor more efficient solutions when tasks can be solved with algorithms with a wider range of time complexities. We used zero-shot chain-of-thought method (Kojima et al., 2022) when prompting these models for solutions. Our choice to focus on zero-shot was deliberate, as adding few-shot examples would introduce a confounding variable—the selection and structure of the examples—which could bias the results and obscure the intrinsic effects of model generalization.

### 4.1 PERFORMANCE ON ID AND OOD DATASET

We begin by evaluating Qwen-1.5 models on their generated ID and OOD datasets, as shown in Fig. 5. Accuracy on ID data consistently increases with model size. Smaller models (1.8B, 4B) show lower and more variable performance, whereas larger models (32B, 72B, 110B) achieve nearly perfect accuracy with reduced variance. OOD performance also improves with model size. However, the gap between ID and OOD accuracy persists, especially in smaller models. Larger models (32B and above) demonstrate significant OOD gains but still show greater variance in performance compared to ID tasks, highlighting the ongoing challenge of generalization.

## 4.2 THE GENERALIZATION VALLEY PHENOMENON

Building on our initial evaluation of ID and OOD performance across different model sizes, we now specifically focus on the performance gap between ID and OOD data: $\max(0, A_{\text{ID}} - A_{\text{OOD}})$, where $A_{\text{ID}}$ and $A_{\text{OOD}}$ denote accuracy on ID and OOD data, respectively. The performance gap between ID and OOD data provides deeper insights into the models' reliance on memorization versus their ability to generalize. As shown in Fig. 6, results for Llama-3.2-3B, Llama-3-8B, Gemma-2-9B, Claude-3-Sonnet, and GPT-4o reveal a non-monotonic relationship between task complexity and the ID-OOD performance gap, which we refer to as *generalization valley*. Specifically, as task complexity increases, the gap widens, reaching a peak where models rely most on memorization, meaning that the model performs well on ID tasks but poorly on OOD tasks.

This peak marks the point of *critical complexity*, at which models struggle to generalize to unseen data, over-relying on memorized patterns from the training distribution. Fig. 6 also shows how models exhibit a clear drop in OOD accuracy as task reaches the critical complexity, which signals that models' ability to generalize is overwhelmed by the complexity of the task, leading to a sharp increase in the performance gap. Beyond this peak, both ID and OOD accuracy decline significantly, and the perfor-

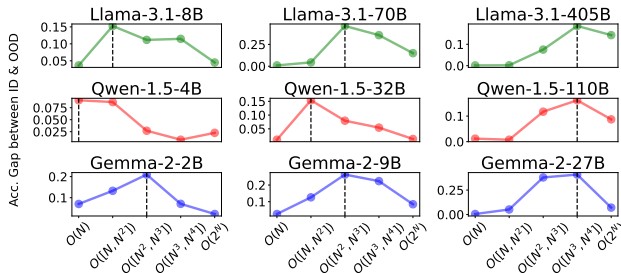

Figure 7: Critical complexity shifts to the right as model size increases for open-sourced LLMs, indicating enhanced generalization capacity.

mance gap stabilizes at a lower level, reflecting the model's failure of relying on either memorization or generalization, showing model's diminished capacity to handle tasks of higher complexity.

In Fig. 7 and Fig. 8, we compute the performance gap between ID and OOD test data, and compare open-sourced and close-sourced models across various sizes. We observe a rightward shift of the critical complexity as models scale up in size. As shown in Fig. 7, larger models, such as Llama-3.1-405B, handle more complex tasks before reaching their peak reliance on memorization. This suggests that increasing model size enhances the model's generalization capacity, enabling better performance on harder tasks before overfitting to ID data becomes a dominant factor. However, even the

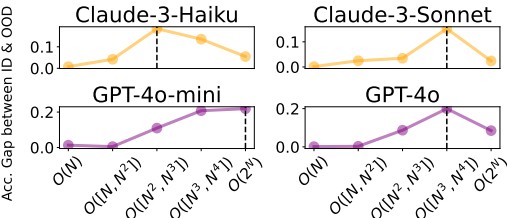

Figure 8: Critical complexity shifts for closed-source models: Sonnet shows higher critical complexity than Haiku, while GPT-4o-mini deviates from this trend.

largest models eventually reach critical complexity at very high task difficulties, suggesting that while scaling delays the reliance on memorization, it does not entirely eliminate it. We also notice that GPT-4o-mini, despite having a smaller size than GPT-4o, exhibits a higher critical complexity (bottom-left in Fig. 8). We hypothesize that GPT-4o-mini might have undergone more aggressive or sophisticated training on a high-quality dataset based on GPT-4o, leading to enhanced generalization capabilities at higher task complexities.

## 4.3 GENERALIZATION SCORE

Based on performance on ID and OOD data, we propose a new metric termed the **Generalization Score** ($S$). This metric is designed to reward models that perform well on OOD data while penalizing those that overfit to ID data. The generalization score is defined as: $S = A_{\text{OOD}} - \max(0, A_{\text{ID}} - A_{\text{OOD}})$, where $A_{\text{ID}}$ represents the accuracy on ID data, and $A_{\text{OOD}}$ denotes the accuracy on OOD data. The score $S$ ranges from $-1$ to $1$, inclusive. The rationale behind this metric is twofold: 1) encouraging high OOD performance, where a higher $A_{\text{OOD}}$ indicates that the model can generalize learned skills to new, unseen data distributions, reflecting genuine reasoning abilities rather than mere memorization. And 2) penalizing over-reliance on memorization: the term $\max(0, A_{\text{ID}} - A_{\text{OOD}})$ measures the performance gap between ID and OOD data. If the model performs significantly better on ID data than on OOD data, this suggests reliance on mem-

orized patterns specific to the training distribution. An ideal LLM reasoner should achieve high accuracy on OOD data ($A_{\text{OOD}}$ close to 1) and exhibit a minimal performance gap between ID and OOD data ($A_{\text{ID}} - A_{\text{OOD}}$ close to 0). This results in a generalization score $S$ approaching 1. Conversely, a model that performs well on ID data but poorly on OOD data—indicating heavy reliance on memorization—would have a larger performance gap, leading to a lower or even negative $S$.

## 4.4 BENCHMARK RESULTS

We evaluate 28 models across different families, versions, sizes, and expertise, including Qwen family (Qwen-1.5-1.8B/4B/7B/14B/32B/72B/110B, Qwen-2-7B, Qwen-2.5-3B/7B, Qwen-2.5-Coder-7B, and Qwen-2.5-Math-7B), LLaMA family (LLaMA-2-7B, LLaMA-3-8B/70B, LLaMA-3.1-8B/70B/405B, and LLaMA-3.2-3B), Gemma family (Gemma-2-2B/9B/27B), GPT family (GPT-4o, GPT-4o-mini, GPT-o1-mini), Claude family (Claude-3-Sonnet/Haiku), and Mistral-7B-v0.3. Unless otherwise specified, all tested models are instruction-tuned models. We test each LLM and report their generalization scores in Table 2.

The results demonstrate that closed-sourced models generally exhibit stronger generalization abilities and achieve higher critical complexity than their open-sourced counterparts. Notably, GPT-4o-mini and o1-mini are the only models that reach the critical complexity of $O(2^N)$, indicating their ability to handle a broad range of highly complex tasks. The o1-mini model, in particular, outperforms all other models across every complexity class, achieving a perfect generalization score of 1 on $O(N)$ tasks and maintaining superior performance even in the most challenging $O(2^N)$ category. Revisiting Fig. 6, the performance gap curve for o1-mini remains almost flat when compared on the same scale as other models, indicating its exceptional generalization abilities across task complexities, with minimal dependence on memorization. More discussions and examples of how o1-mini reasons on SCYLLA's tasks can be found in Appendix F.

Among open-sourced models, Llama-3.1-405B stands out with the strongest performance, achieving near-perfect generalization scores of 0.997 and 0.996 on $O(N)$ and $O([N, N^2])$ tasks, respectively, and exhibiting robust generalization up to $O([N^3, N^4])$ complexity. This suggests that scaling open-source models, such as Llama, can significantly improve their generalization capabilities, approaching the performance of proprietary models at the higher end of the task complexity spectrum. Within 7B-9B models, Qwen-2.5-7B emerges as the top performer, with generalization scores of 0.617 and 0.632 for $O(N)$ and $O([N, N^2])$, respectively, and maintaining strong performance at higher complexity levels, with scores of 0.514 and 0.136 for tasks in the $O([N^2, N^3])$ and $O([N^3, N^4])$ ranges. This positions Qwen-2.5-7B as a leading contender for medium-scale open-sourced models capable of handling diverse levels of task complexity.

We also compare models fine-tuned for specific domains, such as math and code, against their base versions. Qwen2.5-Math-7B and Qwen2.5-Coder-7B outperform the base Qwen2.5-7B on tasks of lower complexity ($O(N)$ and $O([N, N^2])$), but do not necessarily show improved results on tasks of higher complexity. This suggests that domain-specific fine-tuning enhances a model's generalization within the complexity range it was originally effective in, but offers limited gains for tasks that exceed this range.

## 4.5 WHICH COMPLEXITY DOES AN LLM PREFER TO SOLVE A TASK?

To further explore our benchmark, we designed a set of probe tasks to analyze the time complexity that different LLMs employ to solve these tasks. First, we constructed a mapping that correlates time complexity with OOD accuracy. For this analysis, we used Llama-3-8B and Mistral-7B as examples. Initially, we plotted the mapping, represented by the black line in the graph. Next, we tested the accuracy of these two models on three probe tasks: longest common subarray (LCS), longest increasing subsequence (LIS), and longest consecutive elements (LCE). The results are shown in Fig. 9, where green, yellow, and purple lines represent the accuracy for each task. By identifying the intersection points between the accuracy results and the mapping curves, we can infer the complexity

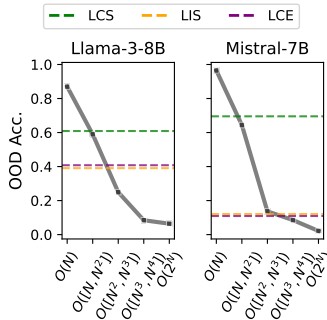

Figure 9: Illustration of correlations between task complexity and OOD accuracy for three probe tasks.

of algorithms used by each LLM based on the corresponding x-coordinate of the intersection points.

Table 2: Generalization scores and critical complexities of 28 LLMs. The highest scores within each complexity range are highlighted in **bold**, and the second-highest scores are underlined.

| Model | $O(N)$ | $O([N, N^2])$ | $O([N^2, N^3])$ | $O([N^3, N^4])$ | $O(2^N)$ | Critical Complexity |
|---|---|---|---|---|---|---|
| **Open-sourced Models** | | | | | | |
| Qwen1.5-1.8B | 0.547 | 0.226 | 0.042 | -0.009 | 0.018 | $O(N)$ |
| Qwen1.5-4B | 0.631 | 0.173 | 0.094 | 0.061 | 0.007 | $O(N)$ |
| Qwen1.5-7B | 0.887 | 0.268 | 0.152 | 0.063 | 0.056 | $O([N, N^2])$ |
| Qwen1.5-14B | 0.849 | 0.568 | 0.075 | 0.025 | 0.037 | $O([N, N^2])$ |
| Qwen1.5-32B | 0.969 | 0.521 | 0.137 | 0.102 | 0.045 | $O([N, N^2])$ |
| Qwen1.5-72B | 0.911 | 0.825 | 0.115 | -0.027 | 0.01 | $O([N^3, N^4])$ |
| Qwen1.5-110B | 0.965 | 0.95 | 0.129 | 0.044 | 0.01 | $O([N^3, N^4])$ |
| Qwen2-7B | 0.986 | 0.824 | 0.065 | 0.081 | 0.019 | $O([N^2, N^3])$ |
| Qwen2.5-3B | 0.934 | 0.81 | 0.251 | 0.181 | 0.079 | $O([N^2, N^3])$ |
| Qwen2.5-7B | 0.617 | 0.632 | 0.514 | 0.122 | 0.136 | $O([N^3, N^4])$ |
| Qwen2.5-Math-7B | 0.971 | 0.874 | 0.591 | 0.135 | 0.044 | $O([N^3, N^4])$ |
| Qwen2.5-Coder-7B | 0.969 | 0.832 | 0.205 | -0.021 | 0.054 | $O([N^3, N^4])$ |
| Llama-2-7b | 0.699 | 0.295 | 0.001 | 0.042 | -0.051 | $O([N, N^2])$ |
| Llama-3-8B | 0.801 | 0.392 | 0.214 | -0.005 | 0.044 | $O([N, N^2])$ |
| Llama-3-70B | 0.9 | 0.882 | -0.109 | -0.218 | -0.05 | $O([N^3, N^4])$ |
| Llama-3.1-8B | 0.667 | 0.603 | 0.047 | 0.025 | -0.003 | $O([N, N^2])$ |
| Llama-3.1-70B | 0.982 | 0.89 | -0.087 | -0.233 | -0.015 | $O([N^2, N^3])$ |
| Llama-3.1-405B | 0.997 | 0.996 | 0.847 | 0.516 | 0.262 | $O([N^3, N^4])$ |
| Llama-3.2-3B | 0.827 | 0.247 | -0.127 | -0.079 | 0.051 | $O([N, N^2])$ |
| Gemma-2-2b | 0.788 | 0.469 | -0.094 | 0.021 | 0.05 | $O([N^2, N^3])$ |
| Gemma-2-9b | 0.948 | 0.601 | -0.124 | -0.026 | -0.065 | $O([N^2, N^3])$ |
| Gemma-2-27b | 0.99 | 0.869 | -0.1 | -0.164 | 0.022 | $O([N^3, N^4])$ |
| Mistral-7B-v0.3 | 0.949 | 0.558 | -0.029 | 0.078 | 0.012 | $O([N^2, N^3])$ |
| **Closed-sourced Models** | | | | | | |
| Claude-3-haiku | 0.984 | 0.844 | 0.019 | 0 | -0.021 | $O([N^2, N^3])$ |
| Claude-3-sonnet | 0.997 | 0.804 | 0.159 | -0.056 | 0.07 | $O([N^3, N^4])$ |
| GPT-4o | 0.997 | 0.992 | 0.787 | 0.457 | 0.238 | $O([N^3, N^4])$ |
| GPT-4o-mini | 0.974 | 0.989 | 0.768 | 0.517 | -0.008 | $O(2^N)$ |
| o1-mini | **1** | **0.998** | **0.979** | **0.949** | **0.68** | $O(2^N)$ |

Interestingly, the yellow and purple lines are nearly identical, indicating that both models use algorithms with similar time complexities. However, these two LLMs apply different time complexities when solving these two tasks. Mistral-7B uses a less efficient algorithm than Llama-3-8B, which is expected given Llama-3-8B's overall superior performance in our benchmark ($O(N^2)$ vs. $O([N^2, N^3])$). For the "Longest Common Subarray" task, although the accuracy differs, the time complexity of the algorithms used by both models is relatively similar, around $O([N, N^2])$. This suggests that either Mistral-7B is performing better than its average, or Llama-3-8B is performing worse than expected.

## 5 CONCLUSION

In this work, we quantitatively study the generalization abilities of LLMs through the development and use of a novel evaluation framework, SCYLLA, which provides a scalable and dynamic evaluation method to disentangle generalization from memorization, allowing us to assess model performance across both ID and OOD data. Our findings highlight a non-monotonic performance gap between ID and OOD data as task complexity increases, a phenomenon we refer to as the *generalization valley*. This behavior is indicative of the balance between memorization and generalization, which peaks at the *critical complexity*. As model size increases, the peak of this generalization valley shifts towards higher task complexity, demonstrating that larger models exhibit enhanced generalization capabilities before over-relying on memorization. Leveraging the insights provided by SCYLLA, we conducted an extensive benchmarking of 28 LLMs, both open-sourced and proprietary, and more clearly show that larger models still face challenges with more complex tasks.

ETHICS STATEMENT

We affirm our commitment to responsible and ethical conduct throughout the study. No human subjects were involved, and the datasets used were either publicly available or generated without personally identifiable information, avoid any kind of privacy invasion. All models we used are either open-sourced or accessible through APIs. We aim to minimize any negative social impact, including biases in model predictions. All methodologies followed best practices for transparency and reproducibility, adhering to standards of research integrity and legal compliance.

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

APPENDIX

# A    ADDITIONAL RELATED WORK

## A.1    SCALING LAWS FOR LLMS

Scaling laws for LLMs have shown predictable improvements in performance as models grow in size, data, and compute. Kaplan et al. (2020) established that model performance scales smoothly with these factors, following power-law relationships. Larger models are more efficient and require fewer samples to reach the same performance level, emphasizing the importance of scaling model size, data, and compute in tandem. Hoffmann et al. (2022) extended this by showing that many models are undertrained relative to their size and should scale model size and data equally to optimize compute. They introduced Chinchilla, a smaller, compute-efficient model that outperformed larger models like GPT-3 on downstream tasks. Additionally, Wei et al. (2022a) explored the emergence of new abilities at larger scales, showing that some capabilities appear only after models surpass certain thresholds. Isik et al. (2024) focused on downstream performance, such as translation, showing that scaling laws apply to transfer learning tasks but are highly sensitive to the alignment between pretraining and downstream datasets.

## A.2    BENCHMARK COMPARISON

We compare SCYLLA with existing dynamic benchmarks for LLM reasoning across several key dimensions: 1) Scalability, 2) Knowledge Dependency, 3) Memorization Awareness, and 4) the Number of Tasks.

- **CLRS-text** (Markeeva et al., 2024) demonstrates partial scalability, as it does not explicitly organize tasks by increasing complexity. Additionally, it evaluates LMs for ID and OOD performance by training them on traces of algorithm execution, a method that is often impractical for large-scale assessments.
- **DyVal** (Zhu et al., 2023) and **NPHardEval** (Fan et al., 2023) increase task difficulty by extending problem length, introducing confounding factors. While NPHardEval explicitly categorizes tasks by complexity classes, both benchmarks offer limited insights into how LLMs handle varying levels of complexity with generalization abilities.
- **LiveBench** (White et al., 2024) is designed primarily for comprehensive evaluation of LLM capabilities, featuring many knowledge-intensive tasks such as competition-level math problems and language comprehension tasks. As a result, it is not well-suited for testing reasoning abilities alone.
- Wu et al. (2023) introduce tasks along with corresponding counterfactual tasks, enabling explicit evaluation of in-distribution (ID) and out-of-distribution (OOD) performance. However, their framework does not address scalability in difficulty, and some tasks require significant domain knowledge to solve.

Table 3: Comparison of SCYLLA with existing dynamic benchmarks for LLM reasoning.

| Benchmark | Scalability | Knowledge Dependency | Memorization Awareness | Number of Tasks |
|---|---|---|---|---|
| CLRS-text | half | low | half | 30 |
| DyVal | half | low | zero | 7 |
| NPHardEval | half | low | zero | 9 |
| LiveBench | zero | high | zero | 18 |
| Wu et al. (2023) | zero | medium | full | 9 |
| Scylla (Ours) | full | low | full | 20 |

# B    VALIDATION OF ID/OOD DATA GENERATION

To demonstrate that our approximation of ID/OOD data is both reasonable and meaningful, we use OLMo-7B-Instruct as an example to examine how the ID/OOD data we obtained overlaps with the

training data (i.e., genuine ID data). OLMo is an open-sourced LM trained on the Dolma and Tulu datasets. With the tool WIMBD (What's In My Big Data) (Elazar et al., 2023), we were able to count the occurrences of each ID and OOD example within the two training datasets of OLMo. We ran the data generation pipeline using OLMo and counted the ID and OOD examples obtained in Dolma and Tulu across our 17 anchor tasks. We then calculated the average occurrences of examples for each task, and the results are presented in the table below.

| Task | # ID in Dolma | # ID in Tulu | # ID in Dolma + Tulu | # OOD in Dolma | # OOD in Tulu | # OOD in Dolma + Tulu |
|---|---|---|---|---|---|---|
| FindMode | 5214.58 | 5.8375 | 5220.4175 | 0 | 0 | 0 |
| TSP | 154.5773 | 1.3801 | 155.9574 | 0 | 0 | 0 |
| TwoSum | 31609.2667 | 19.1446 | 31628.4113 | 0 | 0 | 0 |
| FourSumMultipleTen | 128.3333 | 9.1145 | 137.4479 | 0 | 6.52e-05 | 6.52e-05 |
| RemoveDuplicateNumbers | 53715.7647 | 74.2690 | 53790.0337 | 1.8235 | 0.0106 | 1.8342 |
| FindTopk | 4293.4 | 8.3762 | 4301.7762 | 0 | 0 | 0 |
| ThreeSum | 515.1 | 6.5652 | 521.6652 | 0 | 0 | 0 |
| FourSum | 5 | 14.5031 | 19.5031 | 0 | 0 | 0 |
| FindMinimum | 299740.0811 | 26.8876 | 299766.9687 | 0 | 0 | 0 |
| SubsetSum | 30.266 | 6.4953 | 36.7613 | 0 | 0 | 0 |
| ThreeSumInRange | 293.4615 | 6.8534 | 300.3149 | 0 | 0 | 0 |
| FourSumInRange | 3453.44 | 50.0364 | 3503.4764 | 0 | 0 | 0 |
| SortNumbers | 79879.9512 | 32.2212 | 79912.1724 | 0 | 0 | 0 |
| ThreeSumMultipleTen | 1300.4286 | 2.0465 | 1302.4751 | 0 | 0 | 0 |
| SubsetSumInRange | 32342.39 | 9.9770 | 32352.3670 | 0 | 0 | 0 |
| FindMaximum | 311014.89 | 27.1039 | 311041.9939 | 0 | 0 | 0 |
| SubsetSumMultipleTen | 44.5 | 0.3118 | 44.8118 | 0 | 0 | 0 |
| **Avg** | **48455.0253** | **17.7131** | **48472.7385** | **0.1073** | **0.0006** | **0.1079** |

Table 4: Summary of ID and OOD statistics for tasks across Dolma and Tulu datasets.

From the results, it is evident that for each of the 17 anchor tasks and across both training datasets (Dolma and Tulu), the occurrence of ID examples far exceeds that of OOD examples. Specifically, nearly all OOD example occurrences are zero, except for the tasks "FourSumMultipleTen" and "RemoveDuplicateNumbers". In contrast, the occurrence of ID examples ranges from thousands to tens of thousands, highlighting a substantial disparity in the overlap of examples with the training data. These findings support the validity and meaningfulness of our method for formulating ID and OOD examples.

## C  IMPLEMENTATION DETAILS

### C.1  TASK TITLES, DESCRIPTIONS, AND ANSWER FORMATS

In Table 5, we detail the 20 tasks in SCYLLA, including their titles, descriptions, and example inputs. This information will be used to construct prompts to evaluate LLMs (more details in the next section, §C.2).

| Task | Title | Description | Example Input |
|------|-------|-------------|---------------|
| Find Minimum | finding minimum | Given a list of numbers separated by spaces, find the smallest number. | 48 2 64 29 85 7 41 |
| Find Maximum | finding maximum | Given a list of numbers separated by spaces, find the largest number. | 74 29 63 40 88 |
| Find Mode | finding mode | Given a list of numbers separated by spaces, find the mode of the numbers. | 9 17 25 9 25 9 17 |
| Find TopK | finding top k | Given a list of numbers $l$ separated by spaces and a positive integer $k$, find the $k$th largest number. | $l$ = 100 90 80 70 60 50, $k$ = 3 |
| Two Sum | finding two numbers adding up to a specific sum | Given a list of numbers $l$ separated by spaces and a target value $k$, find two numbers in the list that add up to the target value. | $l$ = 73 41 29 12 55 4, $k$ = 41 |
| Sort Numbers | sorting numbers | Given a list of numbers separated by spaces, sort the numbers in ascending order. | 58 31 74 15 47 3 |
| Remove Duplicate Numbers | removing duplicate numbers | Given a list of numbers separated by spaces, remove duplicate numbers so that every number appears only once, and output the remaining numbers in their original order. | 8 12 45 78 90 23 45 78 |
| Three Sum MultipleTen | finding three numbers adding up to be multiple of 10 | Given a list of numbers separated by spaces, find three numbers in the list that add up to be a multiple of 10. | 7 13 17 24 28 7 37 43 |
| Three Sum In Range | finding three numbers adding up to be in a specific range | Given a list of numbers $l$ separated by spaces and two numbers $a$ and $b$, find three numbers in the list that add up to a value that is in the range $(a, b)$. | $l$ = 7 12 17 22 27 32, $a$ = 71, $b$ = 81 |
| Three Sum | finding three numbers adding up to a specific sum | Given a list of numbers $l$ separated by spaces and a target value $k$, find three numbers in the list that add up to the target value. | $l$ = 75 30 60 45 90 15, $k$ = 105 |
| Four Sum Multiple Ten | finding four numbers adding up to be multiple of 10 | Given a list of numbers separated by spaces, find four numbers in the list that add up to be a multiple of 10. | 1 9 14 16 25 27 |
| Four Sum In Range | finding four numbers adding up to be in a specific range | Given a list of numbers $l$ separated by spaces and two numbers $a$ and $b$, find four numbers in the list that add up to a value that is in the range $(a, b)$. | $l$ = 15 30 45 60 75 90 105 120, $a$ = 330, $b$ = 360 |
| Four Sum | finding four numbers adding up to a specific sum | Given a list of numbers $l$ separated by spaces and a target value $k$, find four numbers in the list that add up to the target value. | $l$ = 4 9 16 23 28 33, $k$ = 62 |
| Subset Sum Multiple Ten | finding a subset adding up to be multiple of 10 | Given a list of numbers separated by spaces, find a subset in the list that adds up to be a multiple of 10. | 2 8 18 28 38 48 |
| Subset Sum In Range | finding a subset adding up to be in a specific range | Given a list of numbers $l$ separated by spaces and two numbers $a$ and $b$, find a subset in the list that adds up to a value that is in the range $(a, b)$. | $l$ = 3 7 11 15 19 23 27 31 35 39 43 47 51, $a$ = 52, $b$ = 56 |
| Subset Sum | finding the subset of numbers adding up to a specific sum | Given a list of numbers $l$ separated by spaces and a target value $k$, find a set of numbers in the list that add up to the target value. | $l$ = 8 14 20 26 32 38 44 50 56, $k$ = 240 |
| TSP | Traveling Salesman Problem (TSP) | Given a list of cities and the distances between each pair of cities, your goal is to find the shortest path that visits every city once and returns to the starting city. The inputs include 1) $n$: the number of cities; 2) $D$: an adjacency matrix of size $n \times n$ where $D_{ij}$ is the distance between city $i$ and city $j$. The output should be a list of integers representing the order of cities to visit. The cities are indexed from 0 to $n-1$. City 0 is always the starting city. | $n$ = 5, $D$ = [[0, 12, 25, 18, 30],12, 0, 15, 22, 20],[25, 15, 0, 28, 35],[18, 22, 28, 0, 17],[30, 20, 35, 17, 0]] |
| Longest Consecutive Elements | finding the longest consecutive elements | Given a list of numbers separated by spaces, return the longest consecutive number sequence in ascending order. A consecutive sequence is a sequence of numbers where each number is exactly 1 greater than the previous number. | 45 12 46 13 14 15 47 48 |
| Longest Increasing Subsequence | finding the longest increasing subsequence | Given a list of numbers separated by spaces, return the longest strictly increasing subsequence. A subsequence is a list that can be derived from another list by deleting some or no elements without changing the order of the remaining elements. | 2 4 3 5 1 7 6 8 0 |
| Longest Common Subarray | finding the longest common subarray | Given two integer arrays $l_1$ and $l_2$, return the longest common subarray that appears in both arrays. A subarray is a contiguous sequence of numbers within an array. | $l_1$ = 7 14 21 28, $l_2$ = 14 21 28 35 |

Table 5: More details of tasks: Titles, descriptions, and example inputs.

## C.2 PROMPTS

For the first part of the data synthesis pipeline, we prompt the LLM to generate test inputs for a given task. We use the following prompt templates for each task type:

> **Prompt Template 1. Generate Number Lists**
>
> Randomly generate number lists as test inputs for testing a program written for the task of {task_title}. The task description is: {task_description}. Enclose each number list by square brackets, e.g. [x1, x2, x3, x4]. Do not generate the corresponding output. Do not use or generate any code. Make sure that the number elements are non-negative integers. Make sure that the number lists are not empty and not too long. Now please generate as many such number lists as possible:

> **Prompt Template 2. Generate Number List Pairs**
>
> Randomly generate pairs of number lists as test inputs for testing a program written for the task of {task_title}. The task description is: {task_description}. Enclose each of the two number list by square brackets and put them in another list, e.g. [[x1, x2], [x3, x4]]. Do not generate the corresponding output. Do not use or generate any code. Make sure that the number elements are non-negative integers. Make sure that the number lists are not empty and not too long. Now please generate as many such pairs of number lists as possible:

> **Prompt Template 3. Generate Matrices**
>
> Randomly generate some adjacency matrices as test inputs for testing a program written for the task of solving the task of {task_title}. The task description is: {task_description}. Enclose each adjacency matrix by square brackets. Make sure the matrix is symmetric, and each element should be a non-negative integer. For example, [[0, x1, x2, x3], [x1, 0, x4, x5], [x2, x4, 0, x6], [x3, x5, x6, 0]] is a valid adjacency matrix. Do not generate the corresponding output. Do not use or generate any code. Make sure that the adjacency matrices have length larger than 3 but not too large. Now please generate as many such adjacency matrices as possible:

To ask for LLMs' solutions, we use the following prompt template (Kojima et al., 2022):

> **Prompt Template 4. Zero-shot Chain-of-Thought Prompting**
>
> Here is a task: {instruction}. Solve the task with the following input: {input}. IMPORTANT: End your response with "The answer is <ANSWER>" where you should fill <ANSWER> with your final answer and must format the final answer obeying the following rules: {answer_format_requirements}. Your response: Let's think step by step.

## D  DETAILS OF DIFFERENT COMPLEXITY TASKS

The anchor tasks are selected for the formulation of our benchmark. Therefore, they would have only one solution of two solutions with adjacent time complexities. The following is detailed information about these tasks, arranged by their time complexity.

### D.1  $O(N)$

- **Find Minimum.** The task Find Minimum is to find the minimum in the given list of numbers. It requires looping over all the numbers in the list. To be specific, we can set a flag number when looping over it. It is set as the first number at the very beginning. Once a number is smaller than the flag number, it would be replaced. Then the final flag number after looping over the entire list would be the result. Therefore, the time complexity would be $O(N)$.

- **Find Maximum.** The task Find Maximum is to find the maximum in the given list of numbers. The solution would be similar to the previous task. The difference is when conducting the re-

placement of the flag number, the current number should be bigger than the flag number.The time complexity would be $O(N)$.

- **Find Mode.** The task Find Mode is to find the mode in the given list of numbers. The solution requires looping over the numbers. It requires a dictionary when looping over the numbers. When it loops through a number, the count of the number would be added by one. After looping, it takes $O(1)$ time to conduct the search to find the mode.The time complexity would be $O(N)$.

## D.2 $O([N, N^2])$

- **Find Topk.** The task Find Topk is to find the $k^{\text{th}}$ largest number in the given list of unsorted numbers. The method with the worst time complexity would be first conducting sorting. The time complexity would vary between $O(N^2)$ and $O(N)$ based on the time complexity of the sorting algorithm. Then it would take $O(1)$ time to look up the kth largest number in the sorted list.The time complexity would be $O(N^2)$. The method with the best time complexity is to use the **quickselect** algorithm. It partitions the array around a pivot recursively. Numbers that are smaller than the pivot go to one side, and elements larger go to the other. If the pivot number is the kth largest number, the recursion ends, and we get the output. The time complexity of this method is $O(N)$.

- **Sort Numbers.** The task Sort Numbers is to sort the given list of numbers. It is one of the most popular tasks among all the coding tasks. The time complexity varies from $O(N)$ to $O(N^2)$ based on the algorithm. For example, bubble sort would have time complexity as $O(N^2)$. Merge sort would have time complexity as $O(N \log N)$. Counting sort would have time complexity of $O(N)$.

- **Two Sum.** The task Two Sum is to find two numbers that add up to be a given number. The worst solution is to conduct all the permutations. In other words, it would require two loops to loop over all the possible outputs. Therefore, the worst time complexity of this task is $O(N^2)$. Besides, the solution with the best time complexity would be using a hash table to look up and find the last number of the solution. Specifically, we will create a hash table (or dictionary) to store numbers in the list, which takes $O(N)$ time. Then by conducting permutations with one loop, we would get all the possible answers without the last number, in other words, the complement as target-the one number. Since it takes only $O(1)$ time to check the dictionary, the time complexity would be $O(N)$.

- **Remove Duplicate Numbers.** The task Remove Duplicate Number is to remove the duplicate numbers in the given list of numbers. The time complexity would range from $O(N)$ to $O(N^2)$. The method with the worst time complexity is to conduct the brute-force approach. It involves checking each element of the list against every other element to see if a duplicate exists. This leads to $O(N^2)$ time complexity since it requires two nested loops. The method with the best time complexity is to use hash table.To remove duplicates efficiently, iterate through the list, adding each element to a set, which automatically discards duplicates. Then, convert the set back to a list.The time complexity would be $O(N)$.

## D.3 $O([N^2, N^3])$

- **Three Sum Multiple Ten.** The task Three Sum Multiple Ten is to find three numbers whose sum is a multiple of ten. The worst solution is to conduct all the permutations. In other words, it would require three loops to loop over all the possible outputs and test whether their sum is in the given range. Therefore, the worst time complexity of this task is $O(N^3)$. The best time complexity solution is realized by building the hash table (dictionary), conducting permutations with two loop, and referring to the hash table. Specifically, we would conduct the hash table based on the remainder of these numbers when divided by ten. Therefore, it would have the same time complexity as $O(N^2)$.

- **Three Sum In Range.** The task category Three Sum In Range is to find three numbers whose sum is in a specific range. The worst solution is to conduct all the permutations. In other words, it would require three loops to loop over all the possible outputs and test whether their sum is in the given range. Therefore, the worst time complexity of this task is $O(N^3)$. The best time complexity algorithm is realized by building the hash table (dictionary), conducting permutations

with N-1 loop, and referring to the hash table. Therefore, it would have the same time complexity as $O(N^2)$.

- **Three Sum.** The task Three Sum is to find three numbers that add up to be a given number. The worst solution is to conduct all the permutations. In other words, it would require three loops to loop over all the possible outputs. Therefore, the worst time complexity of this task is $O(N^3)$. Besides, the solution with the best time complexity would be using a hash table to look up and find the last number of the solution. Specifically, we will create a hash table (or dictionary) to store numbers in the list, which takes $O(N^2)$ time. Then by conducting permutations with one loop, we would get all the possible answers without the last number, in other words, the complement as target-the one number. Since it takes only $O(1)$ time to check the dictionary, the time complexity would be $O(N^2)$.

## D.4 $O([N^3, N^4])$

- **Four Sum Multiple Ten.** The task Four Sum Multiple Ten is to find four numbers whose sum is a multiple of ten. The worst solution is to conduct all the permutations. In other words, it would require four loops to loop over all the possible outputs and test whether their sum is in the given range. Therefore, the worst time complexity of this task is $O(N^4)$. It is realized by building the hash table (dictionary), conducting permutations with three loops, and referring to the hash table. Specifically, we would conduct the hash table based on the remainder of these numbers when divided by ten. Therefore, it would have the same time complexity as $O(N^3)$.

- **Four Sum In Range.** The task Four Sum In Range is to find four numbers whose sum is in a specific range. The worst solution is to conduct all the permutations. In other words, it would require four loops to loop over all the possible outputs and test whether their sum is in the given range. Therefore, the worst time complexity of this task is $O(N^4)$. The best time complexity algorithm is realized by building the hash table (dictionary), conducting permutations with three loop, and referring to the hash table. Therefore, it would have the same time complexity as $O(N^3)$.

- **Four Sum.** The task Four Sum is to find four numbers that add up to be a given number. The worst solution is to conduct all the permutations. In other words, it would require four loops to loop over all the possible outputs. Therefore, the worst time complexity of this task is $O(N^4)$. Besides, the solution with the best time complexity would be using a hash table to look up and find the last number of the solution. Specifically, we will create a hash table (or dictionary) to store numbers in the list, which takes $O(N^3)$ time. Then by conducting permutations with one loop, we would get all the possible answers without the last number, in other words, the complement as target-the one number. Since it takes only $O(1)$ time to check the dictionary, the time complexity would be $O(N^3)$.

## D.5 $O(2^N)$

- **Subset Sum Multiple Ten.** The task Subset Sum Multiple Ten is to find a subset whose sum is a multiple of ten. The solution is to conduct all the permutations. In other words, it would require iterating through all the subsets and check whether its sum is multiple of ten.Therefore, the time complexity is $O(2^N)$.

- **Subset Sum In Range.** The task Subset Sum In range is to find a subset whose sum is the given range. The solution is to conduct all the permutations. In other words, it would require iterating through all the subsets and check whether its sum is in the target range.Therefore, the time complexity is $O(2^N)$.

- **Subset Sum.** The task Subset Sum is to find a subset whose sum is the target sum.The solution is to conduct all the permutations. In other words, it would require iterating through all the subsets and check whether its sum is the target sum.Therefore, the time complexity is $O(2^N)$.

- **TSP.** The task TSP is the famous NP-hard task. The task is to find the shortest route for a salesman to visit each city exactly once and return to the starting city, given n vertices and n*(n-1) /2 distances between each two of them. The time complexity of this task is $O(2^N)$. The method is to generate all the possible routes that cross all the cities and count the distance of these routes. By comparing them, we get the final answers.The time complexity would be $O(2^N)$.

D.6 PROBE TASKS

As discussed in section 4.5, we set some probing tasks that have several different methods with different time complexities.

**Longest Increasing Subsequence.** The task Longest Increasing Subsequence is to find the longest contiguous increasing subarrays given a list of numbers. The time complexity of these methods ranges from $O(N \log N)$ to $O(2^N)$. The worst time complexity approach is a brute-force approach. It involves checking all possible subsequences of the lists. Indeed, it is checking all the subsets of the list. Therefore, the time complexity would be $O(2^N)$. The best time complexity approach is realized by using Binary Search with Dynamic Programming (DP). We first use a dynamic array to store the smallest possible tail of increasing subsequences of different lengths. Additionally, we create an array `prev` of size n, initialized to -1, to store predecessor indices. For each element `nums[i]`, we use binary search to find the appropriate position in `dp` where this element should go. If the element is larger than any element in `dp`, it will be appended. If it can replace an element in `dp`, replace the smallest element that is greater than or equal to it. Update the `prev` array. If `nums[i]` extends a subsequence, record the predecessor in `prev`. After processing all elements, the length of `dp` gives the length of the LIS. Trace back the sequence using the `prev` array, starting from the last element of the LIS. Reverse the sequence, as it is reconstructed from the end to the beginning. For each binary search, it takes $O(\log N)$ time and there are N elements in the array. Therefore, the overall time complexity would be $O(N \log N)$.

**Longest Common Subarrays.** The task Longest Common Subarrays is to find the longest contiguous subarrays of two given arrays. The time complexity would range from $O(N^2)$ to $O(2^N)$. The worst time complexity is realized by the brute-force approach. It is done by checking all possible subarrays of the two given arrays. For each array, it has $2^N$ subarrays if the LLM cannot distinguish whether it is contiguous. Therefore, the time complexity of matching these subarrays one by one requires $O(2^N)$ time.

The best time complexity approach is using dynamic programming. We define a 2D array `dp` with dimensions $(m + 1) \times (n + 1)$, where each entry `dp[i][j]` represents the "LCS" length between the prefixes `text1[0]` and `text2[0]`. We set `dp[0][j] = 0` for all $0 \le j \le n$ and `dp[i][0] = 0` for all $0 \le i \le m$. For the remaining cases where $i > 0$ and $j > 0$, we use the following recurrence relations: If the characters at the current positions match, *i.e.*, `text1[i-1] = text2[j-1]`, the LCS length is `dp[i-1][j-1] + 1`, as we include this matching character in the subsequence. If the characters do not match, *i.e.*, `text1[i-1]` $\neq$ `text2[j-1]`, we take the maximum LCS length by either excluding the character from `text1` or `text2`, so `dp[i][j] = max(dp[i-1][j], dp[i][j-1])`. After filling the table, the value `dp[m][n]` will contain the length of the LCS. Therefore, the time complexity would be $O(N^2)$ because it loops through the whole 2D array, which has a size of $N^2$.

**Longest Consecutive Elements.** The task Longest Consecutive Elements is to find the longest consecutive sequence in the given array. The time complexity ranges from $O(N)$ to $O(2^N)$. The approach with the worst time complexity would be the brute-force method, where we find all subsets of the array and identify the longest consecutive subsequence. Its time complexity is $O(2^N)$. The approach with the best time complexity uses a hashing function. It checks whether N+1, N+2, and so on, exist in the array for all the numbers in the array. By optimizing the search to avoid checking starting points where N-1 exists, the algorithm ensures each number is processed once, resulting in an overall time complexity of $O(N)$.

# E ADDITIONAL RESULTS: FEWSHOT PROMPTING

We conducted additional experiments to explore whether using fewshot-cot prompting affects our findings. We use Gemma-2-2B-Instruct as the testee LLM, and use 3-shot CoT examples that are obtained from Gemma's correct solutions and are excluded from the test set. Each task has its unique 3 shot examples. The results are summarized in the table below.

Key insights include:

| Metric | $O(N)$ | $O([N - N^2])$ | $O([N^2 - N^3])$ | $O([N^3 - N^4])$ | $O(2^N)$ |
|---|---|---|---|---|---|
| **zeroshot-cot ID acc** | 0.928385 | 0.699227 | 0.313481 | 0.120916 | 0.094084 |
| **fewshot-cot ID acc** | 0.989583 | 0.688128 | 0.376901 | 0.169025 | 0.112932 |
| **zeroshot-cot OOD acc** | 0.858073 | 0.601736 | 0.116127 | 0.050155 | 0.072648 |
| **fewshot-cot OOD acc** | 0.927083 | 0.595475 | 0.214492 | 0.060678 | 0.086146 |
| **zeroshot-cot acc gap** | 0.070312 | 0.132647 | **0.209727** | 0.070761 | 0.022633 |
| **fewshot-cot acc gap** | 0.065104 | 0.092773 | **0.162409** | 0.108347 | 0.026786 |

Table 6: Performance metrics across complexity classes for zeroshot and fewshot chain-of-thought (cot) using Gemma-2-2B-Instruct.

1. Fewshot-cot prompting does improve performance across both ID and OOD datasets compared to zeroshot-cot prompting. This is especially evident in lower-complexity tasks (e.g., $O(N)$ and $O([N - N^2])$).

2. While fewshot-cot generally reduces the accuracy gap between ID and OOD tests, the peak performance gap remains significant, particularly for intermediate complexity tasks (e.g., $O([N^2 - N^3])$).

3. Even with fewshot prompting, the generalization valley phenomenon persists, and the critical complexity still remains at the same complexity level compared to zeroshot prompting.

These additional experiments show that few-shot cot prompting can enhance performance, particularly for ID datasets, but the core phenomena observed in our study—such as the generalization valley and critical complexity—remain robust. Therefore, while prompt techniques like few-shot prompting can modulate performance, they do not fundamentally alter the underlying generalization patterns of the models, as revealed by SCYLLA.

## F  A GLIMPSE OF HOW o1-MINI DOES REASONING

Given the strong performance of o1-mini, we are interested to see how o1-mini manages to solve hard tasks even with OOD data. In this section, we look into the outputs of o1-mini when given hard problems.

In comparing how o1-mini and Llama-3.1-405B solve the problem of finding the longest increasing subsequence (Fig. 10), o1-mini exhibits a more streamlined and intuitive approach. o1-mini starts by checking each number sequentially, quickly forming a subsequence, and then making necessary adjustments when the increasing order is broken. Its solution process is straightforward and efficient, leading to the correct result with minimal overhead. On the other hand, Llama-3.1 applies a dynamic programming algorithm, constructing arrays to track subsequence lengths and then backtracking to identify the correct sequence, which is correct but unnecessary for such a simple input. While both models produce the correct answer, o1-mini's method demonstrates stronger generalization and efficiency, solving the problem without requiring the more detailed intermediate steps employed by Llama-3.1. This highlights o1-mini's superior ability to handle the task with fewer computational resources and reduced reliance on more structured, algorithmic approaches.

One notable feature of the o1-mini (OpenAI, 2024) is its ability to perform implicit complex reasoning through what can be termed "reasoning tokens". We test o1-mini on the task of Subsetsum and TSP (Fig. 11). From its output, we observe two types of implicit reasoning. **Type 1**: o1-mini directly generates correct answers with minimal explanation or intermediate steps. For example, in the Subset Sum task, o1-mini is able to quickly identify subsets that satisfy the target sum with no explicit reasoning chain provided. **Type 2**: o1-mini produces more elaborate reasoning steps, but leaves out important intermediate steps. In TSP, o1-mini provides a clearer explanation of its thought process, including steps like evaluating possible routes, but the most critical parts of the reasoning—such as how it prioritizes routes or recognizes patterns—still occur implicitly, without being explicitly outlined in the output.

Since o1-mini conceals its reasoning process, it is difficult to observe exactly how it arrives at solutions. We hypothesize that its strong generalization ability stems from an advanced training tech-

**Task Description of *LongestIncreasingSubsequence*:** Given a list of numbers separated by spaces, return the longest strictly increasing subsequence. A subsequence is a list that can be derived from another list by deleting some or no elements without changing the order of the remaining elements.

**Llama-3.1-405B**

**- Input:** 7015 2930 9379 2135 4033 7591

**- Output:** To solve this task, we can use dynamic programming. The idea is to maintain an array where each element represents the length of the longest increasing subsequence ending at that position.\n\nHere's a step-by-step solution:\n\n1. Initialize an array `dp` of the same length as the input list, with all elements set to 1. This is because a single element is always an increasing subsequence of length 1.\n\n2. Initialize an array `prev` to keep track of the previous element in the longest increasing subsequence ending at each position.\n\n3. Iterate over the input list. For each element, compare it with all previous elements. If the current element is greater than a previous element, it can potentially extend the increasing subsequence ending at that previous element.\n\n4. If the current element can extend the increasing subsequence ending at a previous element, and the length of the resulting subsequence is greater than the current longest increasing subsequence ending at the current position, update `dp` and `prev` accordingly.\n\n5. After iterating over the entire list, find the position with the maximum value in `dp`. This is the end of the longest increasing subsequence.\n\n6. Use the `prev` array to reconstruct the longest increasing subsequence by backtracking from the position with the maximum value in `dp`.\n\nNow, let's apply this algorithm to the given input:\n\nInput: 7015 2930 9379 2135 4033 7591\n\nAfter applying the algorithm, we get:\n\ndp: [1, 1, 2, 2, 3, 4]\nprev: [-1, -1, 1, 1, 3, 4]\n\nThe maximum value in `dp` is 4, which is at the last position. Backtracking from this position using `prev`, we get:\n\n7591 -> 4033 -> 2135 -> 2930\n\nSo, the longest strictly increasing subsequence is [2930, 2135 is not valid as 2135 < 2930, so 2930 -> 4033 -> 7591 and also 7015 -> 9379 is another increasing subsequence, but the longest one is the first one].\n\nThe answer is [2930, 4033, 7591]. (correct)

**o1-mini**

**- Input:** 9722 3098 7007 4981 1976 5899 6356

**- Output:** Given the list of numbers: 9722, 3098, 7007, 4981, 1976, 5899, 6356, we need to find the longest strictly increasing subsequence.\n\n1. **Start with the first number:** 9722\n - Next number is 3098, which is less than 9722. So, we can't include 3098 in the increasing subsequence starting with 9722.\n\n2. **Start with the second number:** 3098\n - Next number is 7007, which is greater than 3098. So, we include 7007.\n - Next number is 4981, which is less than 7007. So, we can't include 4981.\n - Next number is 1976, which is less than 7007.\n - Next number is 5899, which is greater than 7007. So, we include 5899.\n - Next number is 6356, which is greater than 5899. So, we include 6356.\n \n The subsequence here is: 3098, 7007, 5899, 6356\n\n3. **However, notice that 5899 is less than 7007, which breaks the increasing order.** Therefore, we need to adjust our subsequence.\n - Starting again from 3098:\n - 3098 -> 4981 -> 5899 -> 6356\n\n This sequence is strictly increasing.\n\n4. **Check if there's a longer subsequence:**\n - Starting with 1976: 1976, 5899, 6356\n - This is shorter than the previous subsequence.\n\nTherefore, the longest strictly increasing subsequence is [3098, 4981, 5899, 6356].\n\n**The answer is [3098, 4981, 5899, 6356]**. (correct)

Figure 10: Comparison between o1-mini and Llama-3.1-405B in solving the longest increasing subsequence problem. o1-mini demonstrates a more direct and efficient approach, adjusting the sequence dynamically with minimal steps, while Llama-3.1 employs a complex dynamic programming method involving array tracking and backtracking to reach the solution.

**Task Description of *SubsetSum*:** Given a list of numbers $l$ separated by spaces and a target value $k$, find a set of numbers in the list that add up to the target value.

**[Implicit Reasoning Type 1] (~35 out of 256 instances)**

**- Input:** $l$ = 4331 7679 6499 6300 1993 4301 7760, $k$ = 18304
**- Output:** The answer is [7679, 4331, 4301, 1993]. (correct)

**[Implicit Reasoning Type 2] (~15 out of 256 instances)**

**- Input:** $l$ = 3516 9902 8014 3841 3917 5729, $k$ = 21100
**- Output:** We need to find a subset of the numbers `[3516, 9902, 8014, 3841, 3917, 5729]` that sums up to `21100`.\n\nAfter evaluating different combinations, the following set adds up to the target value:\n\n- **3516**\n- **8014**\n- **3841**\n- **5729**\n\n**Calculation:**\n\\[ 3516 + 8014 + 3841 + 5729 = 21100 \\]\n\nThe answer is [3516, 8014, 3841, 5729]. (correct)

**Task Description of *TSP*:** Given a list of cities and the distances between each pair of cities, your task is to find the shortest path that visits every city once and returns to the starting city. The inputs include 1) $n$: the number of cities; 2) $D$: an adjacency matrix of size $n \times n$ where $D_{ij}$ is the distance between city $i$ and city $j$.

**[Implicit Reasoning Type 1] (~25 out of 256 instances)**

**- Input:** $n$ = 5, $D$ =\n\\[\n\\begin{bmatrix}\n0 & 586 & 248 & 306 & 529 \\\\n586 & 0 & 981 & 717 & 464 \\\\n248 & 981 & 0 & 993 & 393 \\\\n306 & 717 & 993 & 0 & 399 \\\\n529 & 464 & 393 & 399 & 0\n\\end{bmatrix}\n\\]
**- Output:** The answer is [0, 2, 4, 1, 3, 0]. (correct)

**[Implicit Reasoning Type 2] (~50 out of 256 instances)**

**- Input:** $n$ = 5, $D$ =\n\\[\n\\begin{bmatrix}\n0 & 994 & 139 & 786 & 469 \\\\n994 & 0 & 736 & 983 & 471 \\\\n139 & 736 & 0 & 453 & 247 \\\\n786 & 983 & 453 & 0 & 712 \\\\n469 & 471 & 247 & 712 & 0\n\\end{bmatrix}\n\\]
**- Output:** To solve the Traveling Salesman Problem (TSP) for the given input with 5 cities, we evaluate all possible routes that start and end at city 0. By calculating the total distance for each permutation of the other cities (1, 2, 3, 4), we identify the route with the shortest distance.\n\nAfter evaluating all 24 possible permutations, the shortest routes found are:\n\n1. **Route**: [0, 2, 3, 1, 4, 0] \n **Total Distance**: 2515\n\n2. **Route**: [0, 4, 1, 3, 2, 0] \n **Total Distance**: 2515\n\nBoth routes have the same minimal total distance of 2515. For consistency, we'll present the first route found.\n\nThe answer is [0, 2, 3, 1, 4, 0]. (correct)

Figure 11: Evaluation of o1-mini on OOD data for SubsetSum and TSP tasks. The model exhibits two distinct types of implicit reasoning, each occurring in a significant portion of cases.

nique that allows the model to allocate a large number of extra tokens for complex reasoning before producing the final response.

