# OpenReview forum: "Quantifying Generalization Complexity for Large Language Models"
_ICLR.cc/2025/Conference — ICLR 2025 Poster_

### Official Review · Reviewer_r16m · 2024-10-30

**Soundness:** 3
**Presentation:** 4
**Contribution:** 2
**Rating:** 5
**Confidence:** 4

**Summary:**

This paper introduces SCYLLA, a dynamic evaluation framework that quantifies the generalization abilities of LLMs by distinguishing between memorization and true generalization across a spectrum of task complexities. SCYLLA assesses models on both in-distribution and out-of-distribution data across various complexities, revealing a "generalization valley". Results from evaluating 28 LLMs highlight the effects of model size on generalization, offering insights into the threshold at which LLMs shift from generalizable to memorized behavior.

**Strengths:**

1. The proposed method is straightforward and accessible, with clearly illustrated figures that enhance understanding.
2. The study discusses a critical issue by evaluating LLMs' generalization capabilities, providing valuable insights into the models' potential and limitations.
3. The experiments are comprehensive and detailed, offering sufficient details to ensure reproducibility.

**Weaknesses:**

1. The study does not examine how different prompt techniques might impact model performance. If a model can handle more complex OOD problems under few-shot prompts, does that indicate it possesses generalization capabilities?
2. The method of generating OOD tasks by replacing numbers may be overly simplistic, potentially limiting its effectiveness. This approach might not produce genuinely OOD questions or represent all OOD types, as the model could easily recognize and substitute numbers. The tasks may be too similar in structure to ID tasks.
3. Some conclusions, such as the presence of the generalization valley, may lack novelty. Since all models perform well on simple tasks but poorly on complex ones, the existence of the valley and the rightward peak shift might be predictable.

**Questions:**

1. Could simple prompt techniques significantly enhance base model performance and reduce the peak in performance gaps?
2. The following study [1] also examines the relationship between OOD challenges and model size and complexity.

[1] ALCUNA: Large Language Models Meet New Knowledge

---

> ### Author Response · Authors · 2024-11-22
> **Response to Reviewer r16m (Part 1)**
>
> > The study does not examine how different prompt techniques might impact model performance. Could simple prompt techniques significantly enhance base model performance and reduce the peak in performance gaps?
>
> We highly appreciate your suggestion to examine the impact of different prompting techniques on model performance. However, our choice to focus on the zeroshot-cot paradigm was **deliberate**. Adding few-shot examples would introduce a confounding variable—the selection and structure of the examples—which could bias the results and obscure the intrinsic effects of model generalization. Thus, the decision was aligned with our goal of isolating and rigorously evaluating the genuine generalization capabilities of LLMs.
>
> In response to your suggestion, we conducted additional experiments incorporating the fewshot-cot method. We use Gemma-2-2B-Instruct as the LLM, and use 3 shot CoT examples that are obtained from Gemma’s correct solutions and are excluded from the test set. Each task has its unique 3 shot examples.
>
> The results are summarized in the table below:
>
> |       metric      |   $O(N)$   | $O([N-N^2])$ | $O([N^2-N^3])$ | $O([N^3-N^4])$ |  $O(2^N)$  |
> |:-----------------:|:--------:|:-------:|:--------:|:--------:|:-------:|
> |  **zeroshot-cot ID acc**     | 0.928385 | 0.699227| 0.313481 | 0.120916 | 0.094084|
> |  **fewshot-cot ID acc**     | 0.989583 | 0.688128| 0.376901 | 0.169025 | 0.112932|
> | **zeroshot-cot OOD acc**     | 0.858073 | 0.601736| 0.116127 | 0.050155 | 0.072648|
> | **fewshot-cot OOD acc**     | 0.927083 | 0.595475| 0.214492 | 0.060678 | 0.086146|
> | **zeroshot-cot acc gap**     | 0.070312 | 0.132647| **0.209727** | 0.070761 | 0.022633|
> | **fewshot-cot acc gap**     | 0.065104 | 0.092773| **0.162409** | 0.108347 | 0.026786|
>
>
> From the results, fewshot-cot prompting does improve performance across both ID and OOD datasets compared to zeroshot-cot prompting. This is especially evident in lower-complexity tasks (e.g., $O(N)$ and $O([N-N^2])$).
> While fewshot-cot generally reduces the accuracy gap between ID and OOD tests, the peak performance gap remains significant, particularly for intermediate complexity tasks (e.g., $O([N^2-N^3])$).
> Even with fewshot prompting, the generalization valley phenomenon persists, and the critical complexity still remains at the same complexity level compared to zeroshot prompting.
>
> These additional experiments support the hypothesis that few-shot cot prompting can enhance performance, particularly for ID datasets, but the core phenomena observed in our study—such as the generalization valley and critical complexity—remain robust. Therefore, while prompt techniques like fewshot can modulate performance, they do not fundamentally alter the underlying generalization patterns of the models, as revealed by SCYLLA.
>
> We hope this addresses your concerns and clarifies our methodological choices. Thank you for your constructive feedback, which has strengthened our analysis.

---

> > ### Author Response · Authors · 2024-11-22
> > **Response to Reviewer r16m (Part 2)**
> >
> > > This approach might not produce genuinely OOD questions or represent all OOD types.
> >
> > Thank you for highlighting the limitations of the ID/OOD definitions.
> >
> > We first want to clarify that our evaluation framework is designed to assess the generalization ability of any LLM with only black-box access. Consequently, obtaining genuine ID or OOD data is inherently impossible. To address this, we propose a workaround by prompting the model to generate ID data.
> >
> > To demonstrate that this approximation is both reasonable and meaningful, we use OLMo-7B-Instruct as an example to examine how the ID/OOD data we obtained overlaps with the training data (i.e., genuine ID data). OLMo is an open-sourced language model trained on the Dolma and Tulu datasets. By utilizing the tool WIMBD (What’s In My Big Data) [1], we were able to count the occurrences of each ID and OOD example within the two training datasets of OLMo.
> >
> > We ran the pipeline proposed in our work using OLMo and counted the ID and OOD examples obtained in Dolma and Tulu across our 17 anchor tasks. We then calculated the average occurrences of examples for each task, and the results are presented in the table below:
> >
> > | Task|**# ID in Dolma**|**# ID in Tulu**|**# ID in Dolma + Tulu**|**# OOD in Dolma**|**# OOD in Tulu**|**# OOD in Dolma + Tulu**|
> > |-------------------------|-------------------|------------------|--------------------------|--------------------|-------------------|---------------------------|
> > | FindMode               | 5214.58          | 5.8375| 5220.4175               | 0                  | 0                 | 0|
> > | TSP| 154.5773         | 1.3801| 155.9574               | 0                  | 0                 | 0|
> > | TwoSum| 31609.2667       | 19.1446          | 31628.4113             | 0                  | 0                 | 0|
> > | FourSumMultipleTen     | 128.3333         | 9.1145           | 137.4479               | 0                  | 6.52e-05| 0.0000652                 |
> > | RemoveDuplicateNumbers | 53715.7647       | 74.2690          | 53790.0337             | 1.8235| 0.0106            | 1.8342                    |
> > | FindTopk| 4293.4           | 8.3762           | 4301.7762| 0                  | 0              | 0                         |
> > | ThreeSum| 515.1            | 6.5652           | 521.6652               | 0                  | 0                 | 0                         |
> > | FourSum | 5| 14.5031          | 19.5031                | 0                  | 0                 | 0                         |
> > | FindMinimum            | 299740.0811      | 26.8876          | 299766.9687            | 0                  | 0                 | 0                         |
> > | SubsetSum| 30.266           | 6.4953           | 36.7613                | 0                  | 0                 | 0|
> > | ThreeSumInRange        | 293.4615         | 6.8534           | 300.3149               | 0                  | 0| 0                         |
> > | FourSumInRange         | 3453.44          | 50.0364          | 3503.4764              | 0                  | 0| 0                         |
> > | SortNumbers| 79879.9512       | 32.2212          | 79912.1724             | 0                  | 0                 | 0                         |
> > | ThreeSumMultipleTen    | 1300.4286        | 2.0465           | 1302.4751              | 0                  | 0                 | 0                         |
> > | SubsetSumInRange       | 32342.39         | 9.9770           | 32352.3670             | 0                  | 0                 | 0                         |
> > | FindMaximum| 311014.89        | 27.1039          | 311041.9939            | 0                  | 0                 | 0                         |
> > | SubsetSumMultipleTen   | 44.5             | 0.3118           | 44.8118                | 0                  | 0                 | 0                         |
> > | **Avg**                | **48455.0253**   | **17.7131**      | **48472.7385**         | **0.1073**         | **0.0006**        | **0.1079**                |
> >
> >
> > From the results, it is evident that for each of the 17 anchor tasks and across both training datasets (Dolma and Tulu), the occurrence of ID examples far exceeds that of OOD examples. Specifically, nearly all OOD example occurrences are zero, except for the tasks “FourSumMultipleTen” and “RemoveDuplicateNumbers”. In contrast, the occurrence of ID examples ranges from thousands to tens of thousands, highlighting a substantial disparity in the overlap of examples with the training data. These findings strongly support the validity and meaningfulness of our method for formulating ID and OOD examples.
> >
> > We hope the above experiment addresses your concerns and demonstrates the effectiveness of our example-generation method. We sincerely appreciate your constructive feedback, which has helped enhance our analysis.
> >
> > [1] Elazar, Y., Bhagia, A., Magnusson, I., Ravichander, A., Schwenk, D., Suhr, A., ... & Dodge, J. (2023). What's In My Big Data?. arXiv preprint arXiv:2310.20707.

---

> > > ### Author Response · Authors · 2024-11-22
> > > **Response to Reviewer r16m (Part 3)**
> > >
> > > > The OOD tasks may be too similar in structure to ID tasks.
> > >
> > > We appreciate your concern and make more clarifications here. We would like to clarify that we do not employ distinct "OOD tasks". Instead, we use the **same tasks** with **different test inputs** for ID and OOD evaluation. This approach is a core principle of our task design. The intuition behind is that if the model genuinely understands how to solve a task (i.e., it has generalized the task’s solution), it should perform well regardless of whether the input is drawn from the in-distribution (ID) or out-of-distribution (OOD) set. By keeping the task structure identical and even the input length identical, and varying only the distribution of input elements, we can isolate and evaluate the model’s generalization ability independently of task-specific learning. This approach ensures a fair and consistent assessment of whether the model's reasoning is task-generalizable or overly reliant on memorized patterns from the training data.
> > >
> > > Thank you for highlighting this, and we hope this explanation provides greater clarity.
> > >
> > >
> > > > The following study [1] also examines the relationship between OOD challenges and model size and complexity.
> > >
> > > Thank you for pointing out a related work. The ALCUNA [1] framework evaluates LLMs’ capability to process new knowledge by generating artificial entities and testing models on knowledge understanding, differentiation, and association. It discovers that LLMs perform poorly on reasoning tasks involving synthetic and internal knowledge, presenting challenges in adapting to novel scenarios.
> > >
> > > Our work, on the other hand, focuses on disentangling generalization from memorization. While ALCUNA emphasizes reasoning with new knowledge, we specifically minimize the impact of knowledge and only test the generalization ability. Moreover, we focus on the dynamics between task complexity, model scales, generalization, and memorization, revealing a non-monotonic relationship between task complexity and reliance on memorization, providing insights into how generalization behaviors evolves as models scale up and task complexity scales up.
> > >
> > > With that being said, ALCUNA is a valuable study on OOD challenges and will be cited in our camera-ready version of the paper to contextualize our findings within broader efforts to evaluate LLM generalization.

---

> > > > ### Comment · Reviewer_r16m · 2024-11-22
> > > >
> > > > Any response on:
> > > > > Some conclusions, such as the presence of the generalization valley, may lack novelty. Since all models perform well on simple tasks but poorly on complex ones, the existence of the valley and the rightward peak shift might be predictable.

---

> ### Author Response · Authors · 2024-11-22
> **Response to Reviewer r16m**
>
> We appreciate your observation regarding the predictability of the generalization valley and critical complexity shift, and we acknowledge that it is intuitive that "simpler tasks are easier and complex tasks are harder". However, we reiterate the novelties in our work and hope that our clarifications address your concerns:
>
> 1. The fact that “simple tasks are easy and complex tasks are hard” does imply the rightward shift of critical complexity, but does not imply the existence of the generalization valley: As tasks become more complex, the reliance on non-generalizable behaviors first increases to a peak (critical complexity) and then decreases, indicating an intermediate point where the model still performs well on ID tasks but poorly on OOD tasks. Without this increasing-decreasing trend, we cannot obtain the critical complexity.
>
> 2. We do not intend to argue that "right shift of critical complexity" is an unexpected phenomenon. Conversely, that "right shift of critical complexity is intuitive since larger models should be able to handle more complex tasks" actually shows that the definition of critical complexity is meaningful and aligns with intuition.
>
> 3. The critical complexity highlights an important threshold beyond which the generalization ability drastically diminishes. This metric allows for a quantitative comparison of generalization capabilities across models of varying sizes and architectures, adding rigor to the evaluation.
>
> 4. Even if the generalization valley may seem predictable in hindsight, formalizing this behavior through systematic experiments across multiple models provides a robust empirical framework. Our work benchmarks this phenomenon quantitatively for the first time, enabling the community to use critical complexity as a measurable indicator of model generalization.
>
> 5. Using algorithmic complexity as a controlled metric to evaluate performance provides further insights into why certain tasks cause a pronounced/small performance gap. This approach also reveals differences in how specific models handle memorization versus generalization, contributing new angles for model evaluation.
>
> In conclusion, while our observations may seem intuitive at a high level, our work introduces quantitative rigor, systematic analysis, and actionable metrics that contribute to the overall understanding of LLM generalization behaviors.

---

> > ### Author Response · Authors · 2024-11-23
> > **Follow-up on our previous response**
> >
> > Thank you once again for your thoughtful review and valuable feedback! As we approach the end of the discussion period, we want to ensure that our previous responses have fully addressed all your concerns. If you have any additional questions or unresolved issues that we can clarify to achieve a better score, please don’t hesitate to let us know. We’re more than happy to assist further!

---

> > > ### Comment · Reviewer_r16m · 2024-11-27
> > >
> > > Thank you for your response and some of my concerns have been addressed. However, I still believe that the conclusions and findings presented in the paper are somewhat predictable and lack novelty. Additionally, the task setting appears to be distant from practical applications, which limits the broader impact of the work.
> > > However, the paper is well-written and clear in its presentation, which is commendable. While I maintain my initial score, I would not object to its acceptance if the other reviewers collectively support it.

---

### Official Review · Reviewer_pHuD · 2024-11-03

**Soundness:** 3
**Presentation:** 3
**Contribution:** 3
**Rating:** 6
**Confidence:** 3

**Summary:**

The paper introduces a novel evaluation framework, SCYLLA, designed to assess the generalization capabilities of large language models (LLMs) across varying task complexities. The framework is scalable, dynamic, knowledge-light, and memorization-aware, addressing limitations of existing evaluation methods.
Through extensive experiments on 20 tasks across 5 complexity levels, the authors identify a non-monotonic relationship between task complexity and the performance gap between in-distribution (ID) and out-of-distribution (OOD) data, termed the "generalization valley."
This reveals a critical complexity threshold where reliance on non-generalizable behavior peaks, indicating the upper bound of LLMs' generalization capabilities. The study shows that as model size increases, this critical complexity shifts to higher task complexities, suggesting larger models can handle more complex tasks before over-relying on memorization. The paper benchmarks 28 LLMs, including both open-sourced and closed-sourced models, providing a robust evaluation and deeper understanding of LLMs' generalization abilities. Additionally, a new metric is proposed to reward models with strong OOD generalization and penalize overfitting to ID data.

**Strengths:**

- The idea that the evaluation dataset should be generated dynamically is interesting and can be useful for promoting the accuracy of the evaluation.
- The performance gap between ID and OOD data is also an interesting and novel indicator.

**Weaknesses:**

- The paragraph of Line 300 is not very clear. "From these generated responses, we extract test examples and designate them as ID test data". How could you *extract* test examples? I thought all the generated examples form a candidate pool for test data. What are "the individual components within these examples"? Given that you generate no less than 10k examples, why only 256 are selected in the end?
- Can "ID/OOD data" selected by one model, i.e., mistral-7b, be regarded as ID/OOD data for another model, e.g., GPT-4?
- The scope of your study is narrowed down to (algorithm and numerical-related) reasoning task. However the claim is a little bit of a wider range and is simply referred to as "generalization" as a whole.
- The discussions (e.g., section 4.4 and 4.5) are mainly observed evidence while lack of more insights. E.g., we can easily induce that larger model size brings better generalization, however, there are some other interesting conclusions you may draw by comparing different open-source models that are fine-tuned on code/math or not. Currently only Qwen2.5-Math/Coder-7B is tested but I believe there are other models like deepseek which has different versions of instruction tuning data.

**Questions:**

See weaknesses

---

> ### Author Response · Authors · 2024-11-22
> **Response to Reviewer pHuD**
>
> > How could you extract test examples? What are "the individual components within these examples"?
>
> We apologize for the confusion in the description. To clarify: Suppose the following is an example of response generated by the model after we query it:
> - Test input 1: [1, 2, 3, 4]
> - Test input 2: [5, 6, 7, 8]
>  - Test input 3: [9, 10, 11, 12]
>
> We extract these test examples using regular expressions, yielding the following outputs: `[[1, 2, 3, 4], [5, 6, 7, 8], [9, 10, 11, 12]]`.
>
> The **individual components** are the numbers within these lists, i.e., `[1, 2, 3, 4, 5, 6, 7, 8, 9, 10, 11, 12]`.
>
> > Given that you generate no less than 10k examples, why only 256 are selected in the end?
>
> We generate over 10k examples to estimate the ID distribution and ensure sufficient diversity for OOD data generation. From this pool, we select 256 examples for testing to balance computational efficiency and statistical robustness. Testing with 256 examples provides reliable and meaningful insights while keeping computational costs manageable.
>
> > Can "ID/OOD data" selected by one model, i.e., mistral-7b, be regarded as ID/OOD data for another model, e.g., GPT-4?
>
> No, ID/OOD data is model-specific. The ID data reflects patterns familiar to the specific model, which varies depending on its training data.
>
> > The scope of your study is narrowed down to (algorithm and numerical-related) reasoning task. However the claim is a little bit of a wider range and is simply referred to as "generalization" as a whole.
>
> We acknowledge that our study focuses on a narrower scope of algorithmic and numerical reasoning tasks, while the term "generalization" may imply a broader range. However, we claim that this narrower focus was intentional, grounded in the need to satisfy the four critical criteria for task design in our study. Algorithmic and numerical tasks were chosen because they meet these criteria effectively:
>
> 1. **Scalability (Quantifiability):**  Tasks must allow for a clear and systematic scaling of complexity. Algorithmic and numerical problems lend themselves to this by leveraging constructs like time complexity. This quantifiability ensures that model performance can be rigorously evaluated as task difficulty increases.
>
> 2. **Dynamic Generation:**  Robust evaluation requires the ability to dynamically generate test instances to mitigate issues such as data contamination. Algorithmic tasks can be easily randomized and synthesized during testing.
>
> 3. **Knowledge-Light:**  The chosen algorithmic tasks require minimal external knowledge, relying primarily on logical or computational reasoning. This ensures that performance differences arise from reasoning capabilities rather than disparities in pre-existing domain knowledge.
>
> 4. **Memorization Awareness:**  By explicitly differentiating between ID and OOD numerical data, algorithmic tasks allow us to disentangle memorization from generalization.
>
> While this narrower focus enables rigorous and controlled experimentation, we agree that future work should expand the scope to include more diverse categories. For example, natural language based reasoning tasks involving semantic understanding; multimodal scenarios integrating visual, textual, and auditory inputs; applications like financial forecasting or medical diagnostics that simulate real-world complexity. These extensions could enable a more comprehensive understanding of generalization across a broader spectrum of tasks and domains.
>
> We will revise our claims to align more precisely with the current scope of the study, emphasizing its focus on algorithmic and numerical reasoning as a foundational framework for evaluating LLM generalization. We will underscore the study's value as a proof-of-concept, while leaving room for future exploration of broader generalization phenomena.
>
> > other interesting conclusions you may draw by comparing different open-source models that are fine-tuned on code/math or not.
>
> We appreciate your suggestion and would like to highlight that we have already conducted a comparative analysis between base models and domain specific fine-tuned counterparts in Section 4.4 (Line 460). Specifically, we observed that Qwen2.5-Math-7B and Qwen2.5-Coder-7B outperform their base version (Qwen2.5-7B) on simpler tasks (e.g., $O(N)$ and $O([N, N^2])$ levels). However, these fine-tuned models did not exhibit significant improvements on higher-complexity tasks (e.g., $O([N^2-N^3])$ and beyond). This indicates that while domain-specific fine-tuning enhances performance within the original scope of complexity, its benefits diminish for tasks requiring higher reasoning skills. This finding underscores the importance of task alignment when evaluating generalization in fine-tuned models. We acknowledge that extending this analysis to other open-source models, such as DeepSeek, would provide additional insights, and we aim to explore these comparisons in future work. Thank you for emphasizing this area of improvement.

---

> > ### Author Response · Authors · 2024-11-23
> > **Follow-up on our previous response**
> >
> > Thank you once again for your thoughtful review and valuable feedback! As we approach the end of the discussion period, we want to ensure that our previous responses have fully addressed all your concerns. If you have any additional questions or unresolved issues that we can clarify, please don’t hesitate to let us know. We’re more than happy to assist further!

---

> > ### Comment · Reviewer_pHuD · 2024-11-26
> > **Response**
> >
> > Thanks for your response. I think this paper presents a solid work and the updates reflect the hard work of the authors.

---

> ### Comment · Area_Chair_ioEC · 2024-11-25
> **[Reminder] Response to Authors**
>
> Dear Reviewer,
>
> As the rebuttal period is drawing to a close, I would appreciate your response to the authors' rebuttal at your earliest convenience.
>
> Best Regards,
>
> Area Chair

---

### Official Review · Reviewer_ZwYr · 2024-11-03

**Soundness:** 4
**Presentation:** 4
**Contribution:** 4
**Rating:** 8
**Confidence:** 4

**Summary:**

This paper presents an analysis of the generalization abilities of LLMs. To conduct this analysis, they have created a framework, SCYLLA, composed of tasks with different levels of algorithmic complexity. They use this framework to quantify the balance between generalization and memorization in LLMs, by comparing the results on in-distribution and out-of-distribution samples. From the accuracy difference of the two, they conclude that there exists a generalization valley and a critical complexity in the modes. They present a Generalization Score to measure this phenomenon. Finally, they use probe tasks to analyze the complexity that LLMs use to solve these problems.

**Strengths:**

- New Evaluation Framework (SCYLLA). They present a new evaluation framework that can be used to measure to evaluate the generalization performance of LLMs.
- They uncover and measure the Generalization Valley Phenomenon. According to the experiments conducted on several models, there seems to be a consistent gap between their ID and OOD evaluations. And this gap has a peak (critical complexity) that shifts to the right as model size increases in open-source models.
-  They conduct a large set of experiments and evaluate a good amount of well-known models.

**Weaknesses:**

- Dependence on approximate ID data. Due to the inability to access pre-training data, it is hard to estimate in-distribution data. The approach presented is a workaround that only focuses in the number generation. They do not present the estimated distribution for the models and their difference with their out-of-distribution data.
- As far as I understand, they focus on tasks that require mathematical operations. They do not show a possible generalization to other kind of tasks. Adding other type of tasks could have benefitted the overall quality of the paper.

**Questions:**

In general it is a great paper and you have performed a great analysis from the results obtained. I have the following questions from the paper:

- When creating the OOD sample, do you create 1 for each family of models or 1 for every single model? I do not think this is made clear in the paper.
- What is the reason to select only problems that require numbers? Could it not be easily added some problem on string manipulation with a very similar strategy?
- What is the takeaway for the gray line in Figure 9, is the goal to present how the OOD accuracy is reduced as the complexity increases? I think this is not highlighted in this case.

---

> ### Author Response · Authors · 2024-11-22
> **Response to Reviewer ZwYr (Part 1)**
>
> > Dependence on approximate ID data.
>
> Thank you for highlighting the limitations of dependence on approximate ID/OOD data.
>
> We first want to clarify that our evaluation framework is designed to assess the generalization ability of any LLM with only black-box access. Consequently, obtaining genuine ID or OOD data is inherently impossible. To address this, we propose a workaround by prompting the model to generate ID data.
>
> To demonstrate that this approximation is both reasonable and meaningful, we use OLMo-7B-Instruct as an example to examine how the ID/OOD data we obtained overlaps with the training data (i.e., genuine ID data). OLMo is an open-sourced language model trained on the Dolma and Tulu datasets. By utilizing the tool WIMBD (What’s In My Big Data) [1], we were able to count the occurrences of each ID and OOD example within the two training datasets of OLMo.
>
> We ran the pipeline proposed in our work using OLMo and counted the ID and OOD examples obtained in Dolma and Tulu across our 17 anchor tasks. We then calculated the average occurrences of examples for each task, and the results are presented in the table below:
>
> | Task|**# ID in Dolma**|**# ID in Tulu**|**# ID in Dolma + Tulu**|**# OOD in Dolma**|**# OOD in Tulu**|**# OOD in Dolma + Tulu**|
> |-------------------------|-------------------|------------------|--------------------------|--------------------|-------------------|---------------------------|
> | FindMode               | 5214.58          | 5.8375| 5220.4175               | 0                  | 0                 | 0|
> | TSP| 154.5773         | 1.3801| 155.9574               | 0                  | 0                 | 0|
> | TwoSum| 31609.2667       | 19.1446          | 31628.4113             | 0                  | 0                 | 0|
> | FourSumMultipleTen     | 128.3333         | 9.1145           | 137.4479               | 0                  | 6.52e-05| 0.0000652                 |
> | RemoveDuplicateNumbers | 53715.7647       | 74.2690          | 53790.0337             | 1.8235| 0.0106            | 1.8342                    |
> | FindTopk| 4293.4           | 8.3762           | 4301.7762              | 0                  | 0              | 0                         |
> | ThreeSum| 515.1            | 6.5652           | 521.6652               | 0                  | 0                 | 0                         |
> | FourSum | 5                | 14.5031          | 19.5031                | 0                  | 0                 | 0                         |
> | FindMinimum            | 299740.0811      | 26.8876          | 299766.9687            | 0                  | 0                 | 0                         |
> | SubsetSum| 30.266           | 6.4953           | 36.7613                | 0                  | 0                 | 0|
> | ThreeSumInRange        | 293.4615         | 6.8534           | 300.3149               | 0                  | 0| 0                         |
> | FourSumInRange         | 3453.44          | 50.0364          | 3503.4764              | 0                  | 0| 0                         |
> | SortNumbers| 79879.9512       | 32.2212          | 79912.1724             | 0                  | 0                 | 0                         |
> | ThreeSumMultipleTen    | 1300.4286        | 2.0465           | 1302.4751              | 0                  | 0                 | 0                         |
> | SubsetSumInRange       | 32342.39         | 9.9770           | 32352.3670             | 0                  | 0                 | 0                         |
> | FindMaximum| 311014.89        | 27.1039          | 311041.9939            | 0                  | 0                 | 0                         |
> | SubsetSumMultipleTen   | 44.5             | 0.3118           | 44.8118                | 0                  | 0                 | 0                         |
> | **Avg**                | **48455.0253**   | **17.7131**      | **48472.7385**         | **0.1073**         | **0.0006**        | **0.1079**                |
>
>
> From the results, it is evident that for each of the 17 anchor tasks and across both training datasets (Dolma and Tulu), the occurrence of ID examples far exceeds that of OOD examples. Specifically, nearly all OOD example occurrences are zero, except for the tasks “FourSumMultipleTen” and “RemoveDuplicateNumbers”. In contrast, the occurrence of ID examples ranges from thousands to tens of thousands, highlighting a substantial disparity in the overlap of examples with the training data. These findings strongly support the validity and meaningfulness of our method for formulating ID and OOD examples.
>
> We hope the above experiment addresses your concerns and demonstrates the effectiveness of our example-generation method. We sincerely appreciate your constructive feedback, which has helped enhance our analysis.
>
> [1] Elazar, Y., Bhagia, A., Magnusson, I., Ravichander, A., Schwenk, D., Suhr, A., ... & Dodge, J. (2023). What's In My Big Data?. arXiv preprint arXiv:2310.20707.

---

> ### Author Response · Authors · 2024-11-22
> **Response to Reviewer ZwYr (Part 2)**
>
> > Limited task types
>
> Thank you very much for your suggestion on incorporating more task types. We claim that this narrower focus was intentional, grounded in the need to satisfy the four critical criteria for task design in our study. Algorithmic and numerical tasks were chosen because they meet these criteria effectively:
>
> 1. **Scalability (Quantifiability):**  Tasks must allow for a clear and systematic scaling of complexity. Algorithmic and numerical problems lend themselves to this by leveraging constructs like time complexity (e.g., $O(N)$, $O(2^N)$). This quantifiability ensures that model performance can be rigorously evaluated as task difficulty increases.
>
> 2. **Dynamic Generation:**  Robust evaluation requires the ability to dynamically generate test instances to mitigate issues such as data contamination. Algorithmic tasks can be easily randomized and synthesized during testing.
>
> 3. **Knowledge-Light:**  The chosen algorithmic tasks require minimal external knowledge, relying primarily on logical or computational reasoning. This ensures that performance differences arise from reasoning capabilities rather than disparities in pre-existing domain knowledge.
>
> 4. **Memorization Awareness:**  By explicitly differentiating between ID and OOD numerical data, algorithmic tasks allow us to disentangle memorization from generalization.
>
> While this narrower focus enables rigorous and controlled experimentation, we agree that future work should expand the scope to include more diverse categories. For example, natural language based reasoning tasks involving semantic understanding and contextual reasoning; multimodal scenarios integrating visual, textual, and auditory inputs; applications like financial forecasting or medical diagnostics that simulate real-world complexity. These extensions could enable a more comprehensive understanding of generalization across a broader spectrum of tasks and domains.
>
>
> > When creating the OOD sample, do you create 1 for each family of models or 1 for every single model? I do not think this is made clear in the paper.
>
> We apologize for the confusion. We create OOD samples for each single model.
>
> > What is the reason to select only problems that require numbers? Could it not be easily added some problem on string manipulation with a very similar strategy?
>
> We appreciate that you bring up the string manipulation tasks as extended task types. We did consider string tasks before, but it is even harder to define in-distribution and out-of-distribution for strings. Plus, the tokenization process adds additional complexity to string-based tasks, which potentially introduces additional confounding factors.
>
>
> > What is the takeaway for the gray line in Figure 9, is the goal to present how the OOD accuracy is reduced as the complexity increases? I think this is not highlighted in this case.
>
> The goal here is to demonstrate “a set of probe tasks to analyze the time complexity that different LLMs employ to solve these tasks”. For the gray line specifically, OOD accuracy does reduce as the complexity increases, but as we have already shown this trend in Figure 5, we did not make more explanations here.

---

> > ### Comment · Reviewer_ZwYr · 2024-11-24
> >
> > Thank you for your comments and extended analysis. It has allowed me to better understand the issue on approximate ID data. I also understand the difficulty to extend the work other type of experiments. The experiments ran seem sufficient for the conducted analysis.

---

### Official Review · Reviewer_r2Jo · 2024-11-06

**Soundness:** 3
**Presentation:** 4
**Contribution:** 3
**Rating:** 6
**Confidence:** 3

**Summary:**

This paper introduces SCYLLA, a framework for evaluating LLMs' generalization abilities across different task complexities. The key findings include:

- Discovery of a "generalization valley" - a non-monotonic relationship between task complexity and the performance gap between in-distribution (ID) and out-of-distribution (OOD) data
- Observation that larger models can handle more complex tasks before reaching their critical complexity - the point where models rely most heavily on memorization
- Benchmark results comparing 28 LLMs, showing closed-source models generally perform better at generalization

**Strengths:**

1. Novel evaluation framework that considers both complexity scaling, generalization, and memorization.
2. Comprehensive evaluation across many modern LLMs
3. Clear methodology for testing generalization capabilities
4. Interesting findings about model scaling and complexity relationships

**Weaknesses:**

1. ID/OOD Definition Issues (Major Issue):


The paper's method of determining ID data by asking the model to generate examples could be problematic
By using model-generated examples to determine what constitutes ID data and simply using complementary numbers for OOD, the paper creates a potentially circular and oversimplified definition of distribution shifts. This approach lacks validation against actual training distributions and may not capture meaningful distribution shifts.

2. Other Issues:


- Limited task types (mostly focused on numerical/algorithmic tasks)
- No ablation studies on different prompt formats or chain-of-thought methods

**Questions:**

N/A

---

> ### Author Response · Authors · 2024-11-22
> **Response to Reviewer r2Jo (Part 1)**
>
> > ID/OOD Definition Issues
>
> Thank you for highlighting the limitations of the ID/OOD definitions.
>
> We first want to clarify that our evaluation framework is designed to assess the generalization ability of any LLM with only black-box access. Consequently, obtaining genuine ID or OOD data is inherently impossible. To address this, we propose a workaround by prompting the model to generate ID data.
>
> To demonstrate that this approximation is both reasonable and meaningful, we use OLMo-7B-Instruct as an example to examine how the ID/OOD data we obtained overlaps with the training data (i.e., genuine ID data). OLMo is an open-sourced language model trained on the Dolma and Tulu datasets. By utilizing the tool WIMBD (What’s In My Big Data) [1], we were able to count the occurrences of each ID and OOD example within the two training datasets of OLMo.
>
> We ran the pipeline proposed in our work using OLMo and counted the ID and OOD examples obtained in Dolma and Tulu across our 17 anchor tasks. We then calculated the average occurrences of examples for each task, and the results are presented in the table below:
>
> | Task|**# ID in Dolma**|**# ID in Tulu**|**# ID in Dolma + Tulu**|**# OOD in Dolma**|**# OOD in Tulu**|**# OOD in Dolma + Tulu**|
> |-------------------------|-------------------|------------------|--------------------------|--------------------|-------------------|---------------------------|
> | FindMode               | 5214.58          | 5.8375| 5220.4175               | 0                  | 0                 | 0|
> | TSP| 154.5773         | 1.3801| 155.9574               | 0                  | 0                 | 0|
> | TwoSum| 31609.2667       | 19.1446          | 31628.4113             | 0                  | 0                 | 0|
> | FourSumMultipleTen     | 128.3333         | 9.1145           | 137.4479               | 0                  | 6.52e-05| 0.0000652                 |
> | RemoveDuplicateNumbers | 53715.7647       | 74.2690          | 53790.0337             | 1.8235| 0.0106            | 1.8342                    |
> | FindTopk| 4293.4           | 8.3762           | 4301.7762              | 0                  | 0              | 0                         |
> | ThreeSum| 515.1            | 6.5652           | 521.6652               | 0                  | 0                 | 0                         |
> | FourSum | 5                | 14.5031          | 19.5031                | 0                  | 0                 | 0                         |
> | FindMinimum            | 299740.0811      | 26.8876          | 299766.9687            | 0                  | 0                 | 0                         |
> | SubsetSum| 30.266           | 6.4953           | 36.7613                | 0                  | 0                 | 0|
> | ThreeSumInRange        | 293.4615         | 6.8534           | 300.3149               | 0                  | 0| 0                         |
> | FourSumInRange         | 3453.44          | 50.0364          | 3503.4764              | 0                  | 0| 0                         |
> | SortNumbers| 79879.9512       | 32.2212          | 79912.1724             | 0                  | 0                 | 0                         |
> | ThreeSumMultipleTen    | 1300.4286        | 2.0465           | 1302.4751              | 0                  | 0                 | 0                         |
> | SubsetSumInRange       | 32342.39         | 9.9770           | 32352.3670             | 0                  | 0                 | 0                         |
> | FindMaximum| 311014.89        | 27.1039          | 311041.9939            | 0                  | 0                 | 0                         |
> | SubsetSumMultipleTen   | 44.5             | 0.3118           | 44.8118                | 0                  | 0                 | 0                         |
> | **Avg**                | **48455.0253**   | **17.7131**      | **48472.7385**         | **0.1073**         | **0.0006**        | **0.1079**                |
>
>
> From the results, it is evident that for each of the 17 anchor tasks and across both training datasets (Dolma and Tulu), the occurrence of ID examples far exceeds that of OOD examples. Specifically, nearly all OOD example occurrences are zero, except for the tasks “FourSumMultipleTen” and “RemoveDuplicateNumbers”. In contrast, the occurrence of ID examples ranges from thousands to tens of thousands, highlighting a substantial disparity in the overlap of examples with the training data. These findings strongly support the validity and meaningfulness of our method for formulating ID and OOD examples.
>
> We hope the above experiment addresses your concerns and demonstrates the effectiveness of our example-generation method. We sincerely appreciate your constructive feedback, which has helped enhance our analysis.
>
> [1] Elazar, Y., Bhagia, A., Magnusson, I., Ravichander, A., Schwenk, D., Suhr, A., ... & Dodge, J. (2023). What's In My Big Data?. arXiv preprint arXiv:2310.20707.

---

> ### Author Response · Authors · 2024-11-22
> **Response to Reviewer r2Jo (Part 2)**
>
> > Limited task types
>
> We claim that this narrower focus was intentional, grounded in the need to satisfy the four critical criteria for task design in our study. Algorithmic and numerical tasks were chosen because they meet these criteria effectively:
>
> 1. **Scalability (Quantifiability):**  Tasks must allow for a clear and systematic scaling of complexity. Algorithmic and numerical problems lend themselves to this by leveraging constructs like time complexity (e.g., $O(N)$, $O(2^N)$). This quantifiability ensures that model performance can be rigorously evaluated as task difficulty increases.
>
> 2. **Dynamic Generation:**  Robust evaluation requires the ability to dynamically generate test instances to mitigate issues such as data contamination. Algorithmic tasks can be easily randomized and synthesized during testing.
>
> 3. **Knowledge-Light:**  The chosen algorithmic tasks require minimal external knowledge, relying primarily on logical or computational reasoning. This ensures that performance differences arise from reasoning capabilities rather than disparities in pre-existing domain knowledge.
>
> 4. **Memorization Awareness:**  By explicitly differentiating between ID and OOD numerical data, algorithmic tasks allow us to disentangle memorization from generalization.
>
> While this narrower focus enables rigorous and controlled experimentation, we agree that future work should expand the scope to include more diverse categories. For example, natural language based reasoning tasks involving semantic understanding and contextual reasoning; multimodal scenarios integrating visual, textual, and auditory inputs; applications like financial forecasting or medical diagnostics that simulate real-world complexity. These extensions could enable a more comprehensive understanding of generalization across a broader spectrum of tasks and domains.
>
>
> > Different prompt formats
>
> We highly appreciate your suggestion to examine the impact of different types of prompting on model performance. However, our choice to focus on the zeroshot-cot paradigm was **deliberate**. Adding few-shot examples would introduce a confounding variable—the selection and structure of the examples—which could bias the results and obscure the intrinsic effects of model generalization. Thus, the decision was aligned with our goal of isolating and rigorously evaluating the genuine generalization capabilities of LLMs.
>
> In response to your suggestion, we conducted additional experiments incorporating the fewshot-cot method. We use Gemma-2-2B-Instruct as the LLM, and use 3 shot CoT examples that are obtained from Gemma’s correct solutions and are excluded from the test set. Each task has its unique 3 shot examples.
>
> The results are summarized in the table below:
>
> |       metric      |   $O(N)$   | $O([N-N^2])$ | $O([N^2-N^3])$ | $O([N^3-N^4])$ |  $O(2^N)$  |
> |:-----------------:|:--------:|:-------:|:--------:|:--------:|:-------:|
> |  **zeroshot-cot ID acc**     | 0.928385 | 0.699227| 0.313481 | 0.120916 | 0.094084|
> |  **fewshot-cot ID acc**     | 0.989583 | 0.688128| 0.376901 | 0.169025 | 0.112932|
> | **zeroshot-cot OOD acc**     | 0.858073 | 0.601736| 0.116127 | 0.050155 | 0.072648|
> | **fewshot-cot OOD acc**     | 0.927083 | 0.595475| 0.214492 | 0.060678 | 0.086146|
> | **zeroshot-cot acc gap**     | 0.070312 | 0.132647| **0.209727** | 0.070761 | 0.022633|
> | **fewshot-cot acc gap**     | 0.065104 | 0.092773| **0.162409** | 0.108347 | 0.026786|
>
>
> From the results, fewshot-cot prompting does improve performance across both ID and OOD datasets compared to zeroshot-cot prompting. This is especially evident in lower-complexity tasks (e.g., $O(N)$ and $O([N-N^2])$).
> While fewshot-cot generally reduces the accuracy gap between ID and OOD tests, the peak performance gap remains significant, particularly for intermediate complexity tasks (e.g., $O([N^2-N^3])$).
> Even with fewshot prompting, the generalization valley phenomenon persists, and the critical complexity still remains at the same complexity level compared to zeroshot prompting.
>
> These additional experiments support the hypothesis that few-shot cot prompting can enhance performance, particularly for ID datasets, but the core phenomena observed in our study—such as the generalization valley and critical complexity—remain robust. Therefore, while prompt techniques like fewshot can modulate performance, they do not fundamentally alter the underlying generalization patterns of the models, as revealed by SCYLLA.
>
> We hope this addresses your concerns and clarifies our methodological choices. Thank you for your constructive feedback, which has strengthened our analysis.

---

> > ### Author Response · Authors · 2024-11-23
> > **Follow-up on our previous response**
> >
> > Thank you once again for your thoughtful review and valuable feedback! As we approach the end of the discussion period, we want to ensure that our previous responses have fully addressed all your concerns. If you have any additional questions or unresolved issues that we can clarify, please don’t hesitate to let us know. We’re more than happy to assist further!

---

> ### Comment · Area_Chair_ioEC · 2024-11-25
> **[Reminder] Response to Authors**
>
> Dear Reviewer,
>
> As the rebuttal period is drawing to a close, I would appreciate your response to the authors' rebuttal at your earliest convenience.
>
> Best Regards,
>
> Area Chair

---

> ### Comment · Reviewer_r2Jo · 2024-12-01
> **Feedback to Rebuttal - 1**
>
> I appreciate the authors conducting additional experiments to address my concerns. My concerns regarding prompts have been resolved. However, I have additional questions about the ID and OOD statistics:
>
> - Why are the numbers of examples in this table shown as float numbers rather than integers?
> - WIMBD uses a very simple inverted index search method, which may limit the statistical accuracy of this table. This limitation should be discussed when presenting these results.
> - Do you have any insights why “FourSumMultipleTen” and “RemoveDuplicateNumbers” has OOD while the others doesn't? like a case study with some examples? I am okay if the authors saying this is just a random case but I am curious if there is something else.
>
> Finally, I am curious whether the data generation process for OOD and ID will be open-sourced. Without this, the contribution of this paper would be somewhat limited.
> Nevertheless, I maintain my positive assessment of this paper.

---

> ### Author Response · Authors · 2024-12-01
> **Response to Reviewer r2Jo**
>
> We highly appreciate your response and feedback, and we are glad to hear that our response resolved your concerns.
>
> Regarding your additional questions:
>
> > Why are the numbers of examples in this table shown as float numbers rather than integers?
>
> This is because we averaged over the occurrences of all examples we obtained for each task on each dataset.
>
> >  WIMBD uses a very simple inverted index search method, which may limit the statistical accuracy of this table. This limitation should be discussed when presenting these results.
>
> Thank you for pointing this out. While WIMBD is simple, we consider that it is the most efficient and straightforward way to validate the statistical meaning. We will discuss the limitations of using it in our newest version.
>
> > Do you have any insights why “FourSumMultipleTen” and “RemoveDuplicateNumbers” has OOD while the others doesn't?
>
> We think that it is more likely just random cases. For "RemoveDuplicateNumbers", we think it may also be because the sequences containing duplicate numbers are more common than others.
>
> > whether the data generation process for OOD and ID will be open-sourced
>
> Yes, the entire pipeline proposed in our work will be open-sourced.

---

### Meta-Review · Area_Chair_ioEC · 2024-12-19

**Metareview:**

This paper introduces SCYLLA, a framework for evaluating the generalization ability of LLMs of varying task complexity. Key findings include the discovery of a "generalization valley" indicating a non-monotonic relationship between task complexity and the ID/OOD performance gap, evidence that larger models can handle more complex tasks before reaching a critical complexity point, and a comprehensive benchmark of several LLMs showing that closed-source models generally perform better in terms of generalization.

The strengths of the paper lie in its novel evaluation framework that considers complexity scaling as well as generalization and memorization. It has comprehensively evaluated modern LLMs, which contributes a lot on the experimental contributions. It has relatively clear methodology, and its strong empirical validation using the OLMo model and training data.

The paper also have some limitations, including its focus primarily on numerical/algorithmic tasks, potential concerns about determining ID/OOD data through model generation, and the somewhat predictable nature of some findings like the generalization valley. The limited ablation studies on different prompt formats and distance from practical applications were also noted as weaknesses. However, these limitations are outweighed by the paper's contributions and thorough methodology.

While some limitations remain regarding task scope and practical applications, they do not diminish this paper's contributions to understanding LLM generalization. Therefore, I recommend accepting this paper.

**Additional Comments On Reviewer Discussion:**

During the review discussion, reviewers raised some concerns that the authors addressed comprehensively during the rebuttal. The validity of using model-generated examples for ID data was questioned first, and authors provided extensive empirical validation using OLMo's training data to demonstrate their method effectively distinguishes ID/OOD examples. Concerns about the narrow focus on numerical/algorithmic tasks were (somewhat) addressed by justifying this choice based on the need for quantifiable complexity and minimal knowledge requirements. The authors also conducted additional few-shot experiments to show their findings remain robust across prompting strategies. While the only negative reviewer, r16m, acknowledged the clear presentation of the paper, they felt that the conclusion was somewhat predictable and questioned its novelty. However, the other three reviewers found the paper to be interesting. Given this feedback, I believe the paper should be accepted.

---

### Decision · Program_Chairs · 2025-01-22

Accept (Poster)